# `EchoRL`: Reinforcement Learning via Rollout Echoing

**Jinhe Bi** [1 2 3 4]  **Aniri** [2 3]  **Minglai Yang** [5]  **Xingcheng Zhou** [6]  **Wenke Huang** [7]  **Sikuan Yan** [2]  **Yujun Wang** [2]
**Zixuan Cao** [2]  **Michael Färber** [8]  **Xun Xiao** [1]  **Volker Tresp** [2 3]  **Yunpu Ma** [2 3 4]

## Abstract

Reinforcement Learning with Verifiable Rewards
is an effective route for post-training to strengthen
the reasoning capability of large language models. However, as training proceeds, the learning
signal can collapse thus makes the training gain
become marginal and ineffective. Specifically,
a growing fraction of prompts' rollouts become
advantage-degenerated: all the self-generated rollouts show verified-success, making the standard
deviation over their rewards be zero; accordingly
each rollout's advantage becomes degenerated
(zero) as well. Given such rollouts' advantages,
the policy-gradient for model optimization eventually vanishes, capping the training performance.
We argue that some of these rollouts still contain valuable learning signals but unfortunately
omitted with the existing RLVR methods. In
this paper, inspired through analyzing the entropy
pattern behind golden trajectories produced by
external expert models, we propose `EchoRL` for
better exploiting the advantage-degenerated rollouts to further improve the training performance.
`EchoRL` is a lightweight module that first identifies an EchoClip from verified-success rollouts
based on their step-level entropy values, and then
feeds this clip back as an auxiliary supervision
signal in the RL objective. Extensive experiments
across 10 benchmarks, 5 LLM backbones, and
4 popular RLVR post-training methods demonstrate that `EchoRL` consistently improves RLVR
post-training with minimal overhead. The code is
available via this repository.

[1]Huawei Heisenberg Research Center [2]LMU Munich [3]Munich
Center for Machine Learning [4]MemAgents Lab [5]University of Arizona [6]TUM [7]College of Computing and Data Science, Nanyang
Technological University, Singapore [8]Technische Universität Dresden. Correspondence to: Jinhe Bi <bijinhe@outlook.com>,
Yunpu Ma <cognitive.yunpu@gmail.com>.

*Proceedings of the 43rd International Conference on Machine
Learning*, Seoul, South Korea. PMLR 306, 2026. Copyright 2026
by the author(s).

## 1. Introduction

Large language models (LLMs) underpin a broad range of
high-stakes applications, from mathematics reasoning, code
generation to scientific problem solving and so on. Their recent gains are closely tied to improved reasoning capability
to sustain coherent, multi-step chains of thought that conclude with correct final answers and lower uncertainty (Bi
et al., 2025b; Tian et al., 2025; Yan et al., 2026b; Wang
et al., 2024b; 2025; 2026c). To achieve this, post-training
has become an essential technique for eliciting and sharpening such reasoning behavior. In particular, Reinforcement
Learning (RL), especially with Verifiable Rewards (RLVR),
provides an effective post-training framework by using feedback signals (Shao et al., 2024; Guo et al., 2025; Yu et al.,
2026). RLVR is proven attractive for improving reasoning
because many benchmarks admit automatically checkable
outcomes, yielding a scalable and reliable supervision. Today, Group Relative Policy Optimization (GRPO) has become the de-facto RLVR post-training framework (Shao
et al., 2024). In a nutshell, given a prompt, GRPO optimizes
the policy by estimating policy-gradient calculated based on
a group-normalized advantage derived with rewards from
a group of sampled rollouts. This group normalization calculation relies on relative ranking: the advantage is derived
by standardizing the rewards of sampled rollouts against
the group's mean and standard deviation. The standardization step converts absolute reward values into relative
advantages.

A key deficiency of this framework is a phenomenon we
term as *advantage degeneration*. As the model's reasoning
capability improves, an increasing number of prompts become easy to answer, resulting in sampled rollout groups
contain verifiable-success rewards, as illustrated on the lefthand-side in Figure 1. As mentioned, because GRPO relies
on variance of the rollout rewards to establish relative rankings, the lack of variation forces the value of each rollout's
advantage to zero. Though generating the rollouts consume
lots of computing resources, this unfortunately silences the
optimizer yielding usable learning signals on these prompts.

Nevertheless, a zero-value advantage derived by the current
approach does not always imply an absence of valuable information that can be utilized for optimization. Our case

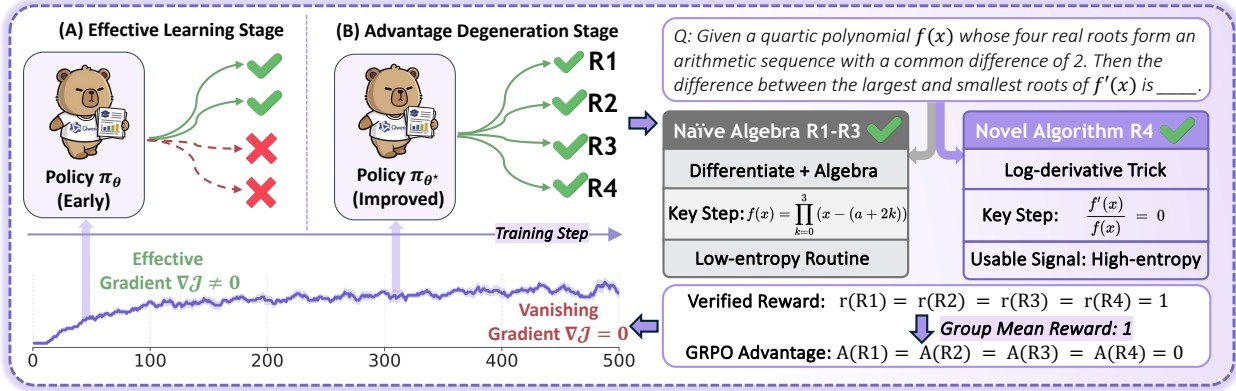

*Figure 1.* **Advantage degeneration and usable learning signals in RLVR. Left:** As training proceeds, groups of verified-success rollouts increasingly suffer advantage degeneration: identical verifiable rewards make the group standard deviation vanish, driving group-relative advantages and policy-gradient updates toward zero. **Right:** For a representative quartic-polynomial prompt during `Qwen2.5-Math-7B` training, verified-success rollouts (R1–R4, all $r = 1$) show low-entropy [Naive Algebra] routines (R1–R3) versus a high-entropy [Novel Algorithm] path (R4) based on the log-derivative trick; standard GRPO nevertheless normalizes them to $A(R1) = \cdots = A(R4) = 0$, discarding this high-entropy usable learning signal.

study (shown on the right-hand-side in Figure 1) says that even if when the outcome rewards are identical (*R1-R4*), some rollouts may still embody qualitatively different reasoning paths. For instance, given the quartic polynomial problem (*Q*), rollouts *R1-R3* solve the problem with brute-force polynomial expansion, but the rollout *R4* employs the *log-derivative trick* to bypass the tedious algebra entirely. Clearly, this specific algorithmic insight represents a high-value usable learning signal that *should* have been reinforced. Yet, because of its identical reward as the others, the standard GRPO will ignore it in the reasoning part, rather treating this valuable reasoning breakthrough as noise, normalizing its advantage to zero and eventually no contribution to the policy-gradient signal.

Not only the standard GRPO, but also its variants suffer the advantage degeneration problem. Prior works typically handle this phenomenon in the following two ways. ❶ One type of strategies simply bypass advantage-degenerated prompts by rejection sampling or dynamic rollout budget allocation. Although this can improve optimization stability, it comes at the cost of reduced data efficiency, low resource utilization, and discarding/wasting potentially informative rollouts; in addition, filtering out non-degenerated groups often requires more rollouts per prompt, which further worsens the time and resource efficiency. ❷ The other type of approaches import golden trajectories from an external expert model to strengthen the learning signal, yet this inherently introduces a heavy dependence on expensive expert models, which might not be always tangible. If left unaddressed, advantage degeneration becomes a bottleneck in RLVR post-training: as training progresses, more rollouts become advantage-degenerated and yield near-zero policy-gradient updates,

slowing down optimization and capping post-training performance. This deficiency in the prior art brings up an open question:

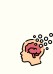 *Is it possible to mine and leverage usable learning signals from generated rollouts even if they receive identical rewards, not simply discarding them or rely on external knowledge?*

**The Present Work:** Our solution `EchoRL` exactly aims to answer the question to exploit those valuable rollouts that cannot be utilized in prior art. However, identifying valuable rollouts is fundamentally challenging because the existing approach calculating a rollout's reward is blind to the *reasoning paths*, i.e., they only focus on the correctness of the final answers, but neglect to distinguish between the quality difference of reasoning paths (Wen et al., 2026; Lightman et al., 2023), as illustrated in our case study.

Motivated by this gap, we develop an empirical diagnostic of where usable learning signals would happen: by contrasting expert golden trajectories with self-generated verified rollouts and analyzing their entropy distributions. We find out that informative supervision systematically clusters around high-entropy decision points in successful reasoning, which often manifests as a sharp entropy peak. This provides us important clues to capture when the model synthesizes an innovative and high-value reasoning. (see Section 4.1 for more details). Thus, instead of relying on an unsubstantiated heuristic, we use *step-level* entropy as the proxy to capture usable learning signals, and `EchoRL` leverages it to mine a critical trajectory clip (i.e., EchoClip) from verified-success rollouts and echo it back as an auxiliary EchoRL update that

can sustain non-vanishing gradients even under advantage degeneration (see Section 4.2 for details). We term this overall two-step procedure—mining EchoClips and echoing them back into the RL objective—*rollout echoing*.

Extensive experiments across four LLM backbones (ranging from 1.5B to 8B parameters) confirm the efficacy of this approach. Our results demonstrate that `EchoRL` consistently improves reasoning capabilities, achieving state-of-the-art gains on nine in-distribution and out-of-distribution benchmarks by effectively mitigating the stagnation caused by advantage degeneration. To sum up, our key contributions can be summarized as follows:

❶ **Bottleneck Identification.** We identify a key deficiency common in RLVR methods—*advantage degeneration*—and show that prior strategies are limited to either discard advantage-degenerated rolluouts or rely on external knowledge from an expert model, leaving a growing fraction of training compute producing near-zero policy-gradient updates while ignoring the usable learning signals in these cases, thereby significantly decreasing post-training efficiency.

❷ **Novel but Practical Solution.** We propose `EchoRL`, a plug-and-play module that modifies the loss function by including an additional term constituted with *EchoClip*s which are captured in verified-success rollouts based on step-level entropy. `EchoRL` guarantees that even if the advantage degenerates to zero, with the new loss function, it can still provide non-trivial gradients.

❸ **Experimental Evaluation.** Extensive experiments across nine benchmarks on five LLM backbones ranging from 1.5B to 8B show that `EchoRL` is **(I)** *high-performing*, improving state-of-the-art RLVR methods by up to 5.61% and 5.04% on in-distribution and out-of-distribution benchmarks , respectively; and **(II)** *resource-friendly*, maintaining comparable or even lower computational cost than mainstream RLVR methods.

## 2. Related Work

Reinforcement learning has firmly established itself as a primary mechanism for enhancing the reasoning capabilities of LLMs, achieving notable success across mathematics, programming, and general problem-solving domains (Team, 2024; 2025; Tian et al., 2025; Li et al., 2026c; Bi et al., 2025a; 2026; Huang et al., 2025; Wan et al., 2025b;a; Yang et al., 2026; Zhang et al., 2023; Peng et al., 2025; Wang et al., 2026b). While earlier post-training pipelines relied on general-purpose policy-gradient algorithms like Trust Region Policy Optimization (TRPO) and Proximal Policy Optimization (PPO) to optimize rollouts against reward feedback (Schulman et al., 2017; 2015), these methods often incur high computational overheads.

More recently, Reinforcement Learning with Verifiable Rewards (RLVR) has emerged as a streamlined alternative, notably through Group Relative Policy Optimization (GRPO). By eliminating the need for an explicit value network (critic) while maintaining competitive performance on reasoning benchmarks, GRPO has become the de-facto standard for scalable post-training (Guo et al., 2025; Shao et al., 2024; Yuan et al., 2026b;a;c; Zhao et al., 2025b;a). Building on the GRPO framework, a growing body of variants has been developed to address specific optimization pathologies and enhance reasoning performance. However, despite their strong empirical performance, current RLVR post-training methods face a notable limitation: as training proceeds, the learning signal can collapse thus makes the training gain become marginal and ineffective. Specifically, a growing fraction of prompts become *advantage-degenerated*: all self-generated rollouts show verified-success, making the standard deviation over their rewards tend to zero; accordingly each rollout's advantage becomes degenerated (zero) as well. As a result, the policy-gradient for model optimization will eventually vanishes, leading the training gain becomes poor. Meanwhile, valid rollouts containing usable learning signals are wasted, leading to significant inefficiency in both data utilization and computational resources—a critical bottleneck given the high cost of RLVR training.

Recent works partially address this limitation through two primary strategies. A first line of work adopts filtering- or sampling-based strategies to avoid training on these weak-signal groups. DAPO (Yu et al., 2026) explicitly detects and filters out advantage-degenerated prompts during training, prioritizing stability over data coverage. Reinforce-Rej (Xiong et al., 2025a) strengthens the signal by employing aggressive rejection sampling. Reinforce-Ada (Xiong et al., 2025b) further refines this direction by dynamically allocating sampling budgets.

A second line of work introduces *auxiliary supervision* to compensate for sparse RL signals, typically by injecting additional golden trajectories. LUFFY (Yan et al., 2026a) augments standard on-policy RLVR with off-policy reasoning traces, allowing the model to learn from additional golden trajectories. UFT (Liu et al., 2026) and SRFT (Fu et al., 2026) both propose a unified training paradigm that integrates Supervised Fine-Tuning (SFT) and RL into a single stage, using mechanisms like dynamic weighting to balance demonstration learning with self-exploration. SEELE (Li et al., 2025b) takes a scaffolding approach, dynamically adjusting problem difficulty via adaptive hints. Last but not least, RelIFT (Ma et al., 2026) employs an interleaved strategy that identifies the model's "hardest" failures during RL and selectively fine-tunes on them using ground-truth solutions to restart progress. Crucially, all aforementioned methods overlook the usable learning signals latently present

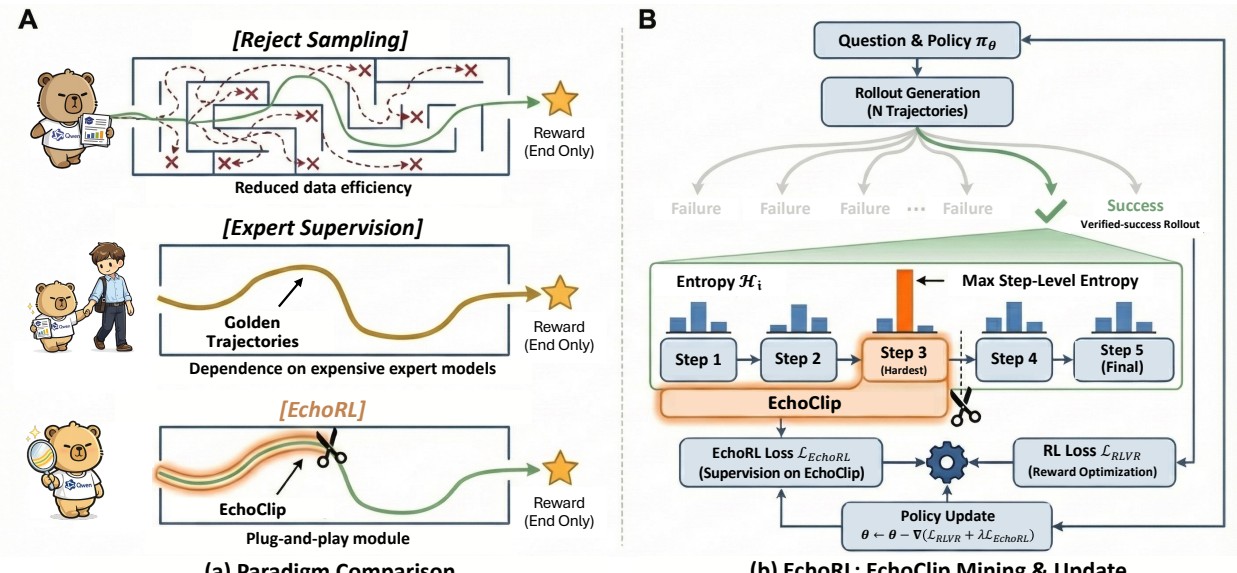

**A** [Reject Sampling]

Reward (End Only)

**Reduced data efficiency**

[Expert Supervision]

**Golden Trajectories**

Reward (End Only)

**Dependence on expensive expert models**

*[EchoRL]*

**EchoClip**

Reward (End Only)

**Plug-and-play module**

**(a) Paradigm Comparison**

**B** Question & Policy $\pi_\theta$

Rollout Generation (N Trajectories)

Failure   Failure   Failure   ⋯   Failure   **Success** Verified-success Rollout

Entropy $\mathcal{H}_i$ ← Max Step-Level Entropy

Step 1 → Step 2 → Step 3 (Hardest) → Step 4 → Step 5 (Final)

**EchoClip**

EchoRL Loss $\mathcal{L}_{EchoRL}$ (Supervision on EchoClip)    RL Loss $\mathcal{L}_{RLVR}$ (Reward Optimization)

Policy Update $\theta \leftarrow \theta - \nabla(\mathcal{L}_{RLVR} + \lambda\mathcal{L}_{EchoRL})$

**(b) EchoRL: EchoClip Mining & Update**

*Figure 2.* **Overview of EchoRL. (a) Paradigm Comparison:** Unlike rejection sampling that discards verified-success rollouts (top) or expert supervision that relies on external golden trajectories (middle), `EchoRL` (bottom) mines usable learning signals directly from the model's own verified rollouts via step-entropy clipping. **(b) EchoRL:** For a verified-success rollout, we calculate step-level entropy to identify the hardest reasoning step (e.g., Step 3). We extract the trajectory prefix ending at this step as an EchoClip. An auxiliary gradient update on this clip ($\mathcal{L}_{\text{EchoRL}}$) is combined with the standard RLVR objective ($\mathcal{L}_{\text{RLVR}}$) using coefficient $\lambda$, sustaining training progress even when advantage degeneration occurs.

within the rollouts of advantage-degenerated prompts. In contrast, `EchoRL` explicitly extracts this usable signal and functions as a plug-and-play module to seamlessly enhance the above frameworks.

## 3. Preliminaries

This section establishes the notation used throughout the paper and briefly reviews the GRPO objective under RLVR. We also introduce a concise definition of *advantage degeneration*, which will be used later to motivate our method and analysis.

**Notation.** Let $q$ denote an input query, and let $\mathcal{D}$ denote a dataset of prompts $(q, a)$ (where $a$ is the reference answer when available). Given a rollout $o = (o_1, \dots, o_{|o|})$ to $q$, its likelihood under $\pi_\theta$ factorizes as

$$\pi_\theta(o \mid q) = \prod_{t=1}^{|o|} \pi_\theta(o_t \mid q, o_{<t}), \quad (1)$$

where $\pi_\theta$ represents the behavior policy of an autoregressive language model parameterized by $\theta$, and $o_{<t} \triangleq (o_1, \dots, o_{t-1})$ and $|o|$ is the number of tokens in $o$.

**Group Relative Policy Optimization (GRPO).** Given a prompt $(q, a)$, GRPO (Shao et al., 2024; Guo et al., 2025) proceeds as follows. First, the behavior policy $\pi_{\theta_{\text{old}}}$ samples a group of $N$ rollouts $\{o^{(i)}\}_{i=1}^N$. Let $r_i$ be the verifiable reward of $o^{(i)}$. GRPO forms group-relative advantages by

normalizing rewards within the group:

$$\hat{A}_i = \frac{r_i - \mu_r(q)}{\sigma_r(q)}, \quad (2)$$

where $\mu_r(q) \triangleq \text{mean}(\{r_j\}_{j=1}^N)$ and $\sigma_r(q) \triangleq \text{std}(\{r_j\}_{j=1}^N)$ are computed from the $N$ rollouts sampled for $q$, and $\hat{A}_i$ is broadcast over time-step $t$ for response $o^{(i)}$.

**Advantage degeneration.** We say a prompt $(q, a)$ is *advantage-degenerated* if the reward standard deviation within its rollout group is (numerically) zero, i.e.,

$$\sigma_r(q) = 0. \quad (3)$$

GRPO framework uses a clipped objective (loss) function built with the group-normalized advantages $A_i$ and a KL regularization term that constrains divergence from a reference model $\pi_{\text{ref}}$. The loss function reads:

$$J_{\text{GRPO}}(\theta) = \mathbb{E}_{\substack{(q,a)\sim\mathcal{D} \\ o^{1:N}\sim\pi_{\theta_{\text{old}}}(\cdot|q)}} \left[ \frac{1}{N} \sum_{i=1}^{N} \frac{1}{|o^{(i)}|} \sum_{t=1}^{|o^{(i)}|} \right.$$
$$\min\left(\rho_{i,t}(\theta)\, \hat{A}_i,\ \text{clip}(\rho_{i,t}(\theta), 1-\epsilon, 1+\epsilon)\, \hat{A}_i\right)$$
$$\left. - \beta\, D_{\text{KL}}(\pi_\theta \,\|\, \pi_{\text{ref}}) \right]. \quad (4)$$

Here $\rho_{i,t}(\theta)$ is the importance ratio between the new and old policy:

$$\rho_{i,t}(\theta) \; = \; \frac{\pi_\theta\left(o_t^{(i)} \mid q, o_{<t}^{(i)}\right)}{\pi_{\theta_{\text{old}}}\left(o_t^{(i)} \mid q, o_{<t}^{(i)}\right)}. \quad (5)$$

Eq. (4) makes explicit that the GRPO update is driven by a policy-gradient term weighted by the group-normalized advantages $\hat{A}_i$. Concretely, the gradient of the clipped surrogate has the form $\nabla_\theta J_{\text{GRPO}}(\theta) \propto \mathbb{E}\left[\sum_{i,t} \nabla_\theta \log \pi_\theta\left(o_t^{(i)} \mid q, o_{<t}^{(i)}\right) \cdot w_{i,t}(\theta) \cdot \hat{A}_i\right]$ for some bounded weight $w_{i,t}(\theta)$ induced by clipping. When advantage degeneration holds (Eq. (3)), we have $r_i = \mu_r(q)$ and thus $\hat{A}_i = (r_i - \mu_r(q))/\sigma_r(q) = 0$ for all $i$, which multiplicatively suppresses the entire policy-gradient signal.

# 4. EchoRL

This section outlines the workflow of `EchoRL`, as illustrated in Figure 2 and Algorithm 1. `EchoRL` is designed to mine and exploit usable learning signals from advantage-degenerated prompts via a two-phase procedure that we call *rollout echoing*, wherein (1) **EchoClip Mining** first identifies a critical trajectory clip (the EchoClip) from a verified-success rollout based on step-level entropy; and (2) **EchoRL Update** then echoes this mined clip back as an auxiliary gradient signal to sustain training progress when standard policy gradients vanish.

## 4.1. Entropy-Based Motivation

To pinpoint usable learning signals in the rollouts generated for a prompt, we start by analyzing *expert model* behavior and ask what distinguishes an external golden trajectory from a self-generated verified-success rollout. We will see that a generic pattern for valuable trajectory clips.

As shown in Figure 3a, the entropy distribution of golden trajectories is systematically higher than that of self-generated rollouts. This implies that useful supervision tends to concentrate near points of high predictive uncertainty. Recall the answer (R4) in our case study, as shown in Figure 3b, R4 exhibits substantially higher entropy than other regions, precisely when the model synthesizes a novel algorithm.

To validate this entropy pattern and quantify its importance, we use entropy as a proxy metric (Wang et al., 2026a; Bi et al., 2025b; Chen et al., 2025; Rong et al., 2026; Li et al., 2026a). Since single-token predictive entropy can be dominated by short-lived lexical fluctuations (e.g., punctuation) and is therefore noisy (Chen et al., 2026; Fomicheva et al., 2020), we aggregate token-level entropy within each reasoning step by averaging to obtain a smoother, step-level uncertainty signal. We then conduct an ablation study on the OpenR1-Math 45k subset (Yan et al., 2026a) to test whether high step-level entropy marks critical reasoning steps. For

each reasoning trajectory, we progressively remove steps based on step-level entropy: *high-entropy* removal discards steps from highest to lowest entropy, *low-entropy* removal discards from lowest to highest, and *random* removal discards steps randomly. As shown in Figure 3c, when only a small portion of steps is removed, high-entropy removal causes substantial accuracy degradation, while low-entropy and random removal require much higher removal ratios to achieve similar performance drops. This establishes that high step-level entropy marks the most critical parts of reasoning trajectories—the usable learning signals. These results support our working hypothesis that high *step-level* entropy can reliably mark usable learning signals in rollouts and motivate using it as the basis for mining *EchoClips* from the verified-success rollouts, which will be introduced next.

## 4.2. EchoClip Mining

Building on this motivation, the mining process extracts the "hardest" correct reasoning clip. Given a prompt $q$, the model first samples a group of rollouts, and we filter for the subset of verified-success rollouts $\mathcal{V} = \{o \mid r(o) = 1\}$. For each verified rollout $o \in \mathcal{V}$, we decompose the trajectory into a sequence of discrete reasoning steps $(s_1, s_2, \dots, s_M)$ using natural delimiters (e.g., \n, see Appendix D).

Formally, let $H_\theta(x)$ be the entropy of the policy's distribution for token $x$. The step-level entropy $\bar{H}(s_j)$ is defined as:

$$\bar{H}(s_j) = \frac{1}{|s_j|} \sum_{x \in s_j} H_\theta(x \mid q, o_{<x}). \quad (6)$$

We identify the single most critical step $s^\star$ that exhibits the maximum entropy *across all* verified rollouts for the current prompt:

$$s^\star = \underset{s \in \text{Steps}(\mathcal{V})}{\arg\max} \bar{H}(s). \quad (7)$$

Let $o^\star \in \mathcal{V}$ be the parent rollout containing $s^\star$. The unique EchoClip $o_{\text{echo}}$ is defined as the prefix of $o^\star$ ending at $s^\star$:

$$o_{\text{echo}} = \text{Prefix}(o^\star, s^\star). \quad (8)$$

This clip captures the model's successful navigation through its most hesitant decision point among all successful attempts, which gives the *EchoClip*.

## 4.3. EchoRL Update

To exploit the mined signal, `EchoRL` introduces a plug-and-play auxiliary module that adapts to current RLVR algorithms (e.g., GRPO, DAPO (Yu et al., 2026), LUFFY (Yan et al., 2026a)). We augment the standard RLVR objective with an auxiliary term $J_{\text{EchoRL}}(\theta)$ applied to the identified EchoClip in one training batch update:

$$J_{\text{EchoRL}}(\theta) = -\frac{1}{L} \sum_{t=1}^{L} \log \pi_\theta((o_{\text{echo}})_t \mid q, (o_{\text{echo}})_{<t}), \quad (9)$$

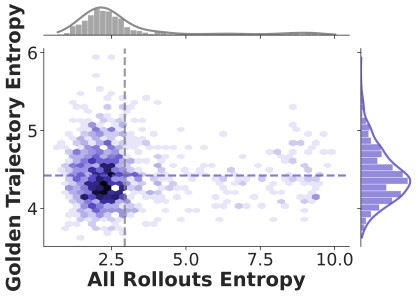

*(a)* Golden trajectories have higher entropy.

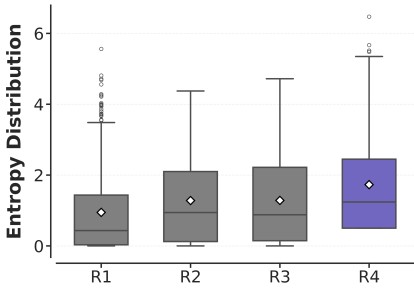

*(b)* R4 exhibits higher entropy than other regions.

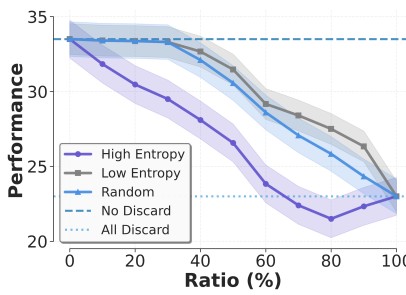

*(c)* High-entropy steps drive performance.

*Figure 3.* **Entropy pattern of usable learning signals in RLVR. (a)** Comparing per-prompt rollouts, external golden trajectories concentrate at higher entropy than regular rollouts from the current policy, indicating that successful expert reasoning often traverses higher-entropy regions. **(b)** Entropy distribution varies across different regions of the generated response: R4 exhibits substantially higher entropy than regions R1–R3. **(c)** We progressively remove reasoning steps based on step-level entropy (high-entropy: removing from highest to lowest; low-entropy: removing from lowest to highest; random: removing randomly). Discarding high-entropy steps causes substantial accuracy degradation even at low removal ratios, while low-entropy and random removal require higher ratios to achieve similar degradation, demonstrating that high step-level entropy marks usable learning signals in reasoning trajectories.

where $L = |o_{\text{echo}}|$ is the token length of the EchoClip. This term is integrated into the total objective:

$$J(\theta) = J_{\text{RLVR}}(\theta) + \lambda J_{\text{EchoRL}}(\theta), \qquad (10)$$

where $\lambda$ is a hyperparameter balancing the RL signal and the auxiliary supervision.

Now we explain why this augmented loss function will mitigate/eliminate advantage degeneration. When all rollouts in a group are successful, the reward variance collapses ($\sigma_r \to 0$), causing the GRPO gradient to vanish. However, $J_{\text{EchoRL}}$ remains active for verified rollouts, providing a stable, dense gradient signal that reinforces the robust resolution of high-uncertainty steps.

## 5. Experiment

In this section, we conduct extensive experiments to answer the following research questions: (**RQ1**) How does `EchoRL` perform when integrated into existing RLVR post-training methods (e.g., GRPO and its variants)? (**RQ2**) Does `EchoRL` introduce significant resource overhead? (**RQ3**) How sensitive is `EchoRL` to its key components and hyperparameters?

### 5.1. Experiment Setup

**Datasets and Benchmarks.** To thoroughly evaluate the effectiveness of `EchoRL`, we adopt 10 widely used benchmarks in challenging reasoning domains. Six are in-distribution math benchmarks: AIME 2024/2025, AMC, MATH-500 (Hendrycks et al., 2021), Minerva (Lewkowycz et al., 2022), and OlympiadBench (He et al., 2024). To assess generalization, we include three out-of-distribution tasks: ARC-c (Clark et al., 2018), GPQA-Diamond (Rein et al., 2024), and MMLU-Pro (Wang et al., 2024a). For AIME 2024, AIME 2025, and AMC, we report avg@32 due to the small test sets; for the remaining benchmarks, we report pass@1. To further validate generalization beyond

textual scenarios, we additionally evaluate on the spatial reasoning benchmark Geometry3K (Lu et al., 2021).

**Baselines.** We evaluate `EchoRL` by integrating it into four representative RLVR post-training methods: GRPO (Shao et al., 2024), DAPO (Yu et al., 2026), LUFFY (Yan et al., 2026a), and UFT (Liu et al., 2026). We also report results from prior work, including: (1) SFT on the OpenR1-Math-46k-8192 (Yan et al., 2026a) dataset; (2) SFT with a KL-divergence constraint incorporated into the loss (SFT-KL); and RL baselines: (3) SimpleRL-Zero (Zeng et al., 2025), applying GRPO to approximately 24k mathematical samples from GSM8K and MATH; (4) OpenReasoner-Zero (Hu et al., 2026), a PPO-based approach trained on 129k multi-source samples including AIME; and (5) PRIME-Zero (Cui et al., 2025), conducting policy rollouts on 150k Numina-Math queries with implicit process rewards and final labels.

**Models.** We consider 5 representative LLMs ranging from 1.5B to 8B parameters from the `Qwen2.5` (Qwen et al., 2025) and `LLaMA-3.1` (Team, 2024) families, and train them on the OpenR1-Math 45k (Yan et al., 2026a). Moreover, we include a multimodal model, `Qwen2.5-VL` (Bai et al., 2025), and train it on Geometry3K (Lu et al., 2021).

**Parameter Configurations.** We implement `EchoRL` in `verl`[1], and in `EasyR1`[2] for the multimodal setting. Following previous work (Liu et al., 2026; Li et al., 2025b;a), we set $\lambda$ to 0.001 to balance the scale of the auxiliary loss. We use a rollout batch size of 128 and an update batch size of 64. During the rollout generation stage, we sample 8 responses per on-policy question. We shuffle multiple-choice options to avoid contamination. For evaluation, the temperature is set to 0.6. All remaining training and evaluation details are provided in Appendix G.

---

[1] https://github.com/volcengine/verl
[2] https://github.com/hiyouga/EasyR1

*Table 1.* Overall in-distribution (ID) and out-of-distribution (OOD) performance on Qwen2.5-Math-7B.

| Model / Method | In-Distribution Performance | | | | | | | Out-of-Distribution Performance | | | |
|---|---|---|---|---|---|---|---|---|---|---|---|
| | AIME24 | AIME25 | AMC | MATH-500 | Minerva | Olympiad | Avg. | ARC-c | GPQA* | MMLU-Pro | Avg. |
| Qwen2.5-Math-7B | 11.4 | 4.9 | 31.3 | 43.6 | 7.4 | 15.6 | 19.0 | 18.2 | 11.1 | 16.9 | 15.4 |
| Qwen2.5-Math-7B-Instruct | 12.9 | 10.2 | 48.5 | 80.4 | 32.7 | 41.0 | 37.6 | 70.3 | 24.7 | 34.1 | 43.0 |
| *Previous Zero RLVR Methods* | | | | | | | | | | | |
| PRIME-Zero | 17.0 | 12.8 | 54.0 | 81.4 | 39.0 | 40.3 | 40.7 | 73.3 | 18.2 | 32.7 | 41.4 |
| SimpleRL-Zero | 27.0 | 6.8 | 54.9 | 76.0 | 25.0 | 34.7 | 37.4 | 30.2 | 23.2 | 34.5 | 29.3 |
| OpenReasoner-Zero | 16.5 | 15.0 | 52.1 | 82.4 | 33.1 | 47.1 | 41.0 | 66.2 | 29.8 | 58.7 | 51.6 |
| *Supervised Learning Methods* | | | | | | | | | | | |
| SFT | 22.2 | 22.3 | 52.8 | 82.6 | 40.8 | 43.7 | 44.1 | 75.2 | 24.7 | 42.7 | 47.5 |
| SFT-KL | 12.4 | 10.8 | 47.1 | 68.4 | 27.5 | 36.3 | 33.8 | 34.1 | 23.2 | 31.5 | 29.6 |
| *Expert-Supervised RLVR with* `EchoRL` | | | | | | | | | | | |
| LUFFY | 29.4 | 23.1 | 65.6 | 87.6 | 37.5 | 57.2 | 50.1 | 80.5 | 39.9 | 53.0 | 57.8 |
| ↪ + `EchoRL` | 33.4 | 25.7 | 67.5 | 88.9 | 39.0 | 55.1 | **51.9** | 83.6 | 45.3 | 54.1 | **61.0** |
| UFT | 24.8 | 18.1 | 60.5 | 82.6 | 40.1 | 47.8 | 45.7 | 82.2 | 38.9 | 49.6 | 56.9 |
| ↪ + `EchoRL` | 27.0 | 21.3 | 62.0 | 84.4 | 40.8 | 49.6 | **47.6** | 82.7 | 43.4 | 53.5 | **59.9** |
| *On-Policy RLVR with* `EchoRL` | | | | | | | | | | | |
| GRPO | 25.8 | 16.4 | 61.2 | 80.4 | 39.7 | 43.7 | 44.5 | 81.8 | 39.9 | 45.2 | 55.6 |
| ↪ + `EchoRL` | 24.9 | 22.3 | 62.7 | 81.4 | 41.2 | 49.3 | **47.0** | 84.7 | 37.4 | 53.0 | **58.4** |
| DAPO | 29.9 | 18.8 | 63.2 | 86.6 | 40.4 | 48.6 | 47.9 | 82.1 | 38.9 | 51.8 | 57.6 |
| ↪ + `EchoRL` | 33.4 | 22.6 | 62.9 | 88.4 | 40.8 | 50.4 | **49.9** | 85.2 | 42.9 | 54.7 | **60.9** |

## 5.2. Main Results (RQ1)

Table 1 and Figures 4 and 5 comprehensively report the performance of seven RLVR methods across three LLM backbones. We summarize the key observations as follows:

**Takeaway ❶:** `EchoRL` **delivers consistent improvements over various RLVR baselines across diverse architectures and domains.** Our experiments demonstrate that `EchoRL` acts as a universal performance booster, effectively improving not only GRPO but also other representative RLVR methods such as DAPO, LUFFY, and UFT. Crucially, these gains are robust across different model families (covering both `Qwen` and `LLaMA`) and scales (ranging from 1.5B to 8B), and extend seamlessly from standard in-distribution math tasks to out-of-distribution generalization and multimodal spatial reasoning.

**Takeaway ❷: The benefits of** `EchoRL` **are more significant to larger model scales.** We observe a distinct trend where the relative improvement of `EchoRL` over baselines increases as the model size grows (e.g., from 1.5B to 8B). In post-training, verified-success rollouts appear much earlier with larger LLMs and/or handling easier tasks. Therefore, advantage degeneration to larger models also happens much earlier. `EchoRL` effectively converts these verified-success rollouts but discarded by standard GRPO methods into dense, usable learning signals, thereby maximizing data efficiency precisely in the regime where baselines stagnate.

**Takeaway ❸:** `EchoRL` **effectively mitigates entropy col-**

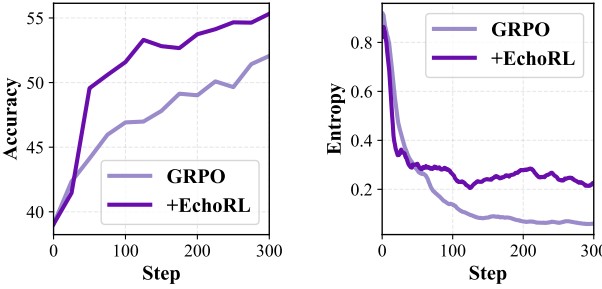

*Figure 4.* **Training Dynamics Analysis. Left:** Accuracy curve of `Qwen2.5-VL-7B` on Geometry3K, confirming `EchoRL`'s generalization to multimodal tasks. **Right:** Step-level entropy evolution on `Qwen2.5-Math-7B`. `EchoRL` maintains significantly higher entropy levels throughout training, indicating sustained exploration capability compared to the rapid collapse observed in GRPO.

**lapse and sustains continuous learning.** Analyzing the training dynamics reveals that while standard GRPO suffers from rapid entropy decay—a signature of the model converging to a narrow set of reasoning paths—`EchoRL` maintains a healthy level of step-level entropy throughout training. This sustained entropy correlates strongly with continuous performance growth, indicating that the auxiliary EchoClip supervision prevents the optimizer from prematurely converging to local optima and provides a persistent learning signal even when the primary reward signal degenerates.

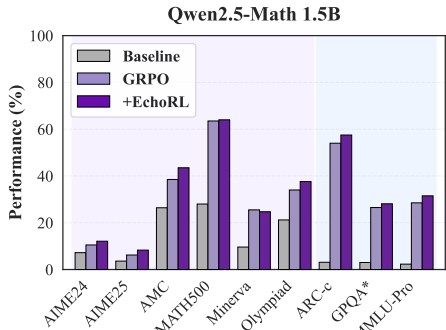
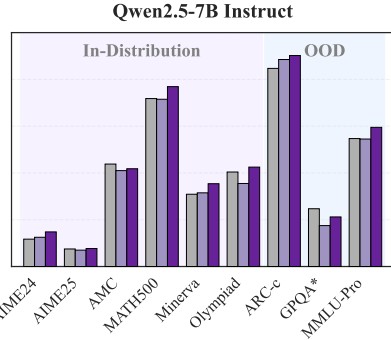
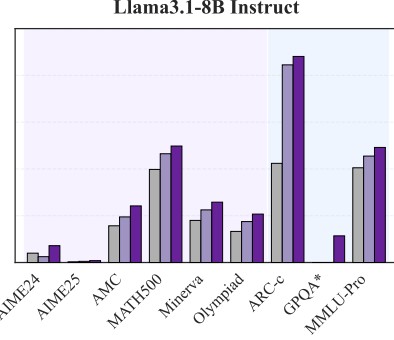

*Figure 5.* **Universality across Model Scales and Families.** To demonstrate the generalizability of `EchoRL` beyond the primary `Qwen2.5-Math-7B` benchmark, we evaluate it across three distinct backbones: the smaller specialized `Qwen2.5-Math-1.5B`, and the general-purpose `Qwen2.5-7B-Instruct` and `Llama3.1-8B-Instruct`. The consistent improvements across all settings verify that our approach is robust to variations in model size, architecture, and pre-training focus.

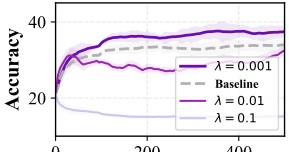

| Method | ID Avg. | OOD Avg. |
|---|---|---|
| GRPO | $44.6 \pm 0.4$ | $55.4 \pm 0.4$ |
| + EchoRL | $46.8 \pm 0.3$ | $58.6 \pm 0.6$ |
| DAPO | $47.7 \pm 0.3$ | $58.0 \pm 0.4$ |
| + EchoRL | $49.6 \pm 0.3$ | $61.1 \pm 0.7$ |

*Figure 6.* **Framework Analysis. Left:** Ablation study on auxiliary loss coefficient $\lambda$, indicating $\lambda = 0.001$ as the optimal balance between exploration and exploitation. **Right:** Sensitivity analysis on `Qwen2.5-Math-7B`, confirming `EchoRL`'s stability across runs and compatibility with RLVR methods.

### 5.3. Cost Analysis (RQ2)

To evaluate the efficiency of `EchoRL`, we analyze its computational cost profile (visualized in Figure 9).

**Takeaway ❹: `EchoRL` is a "free lunch" upgrade with negligible system overhead.** Our empirical profiling confirms that `EchoRL` introduces virtually zero additional latency or memory usage compared to standard GRPO. This efficiency stems from our unified optimization strategy: instead of performing separate updates, we merge the auxiliary EchoClip loss with the primary objective and execute a single, unified backward pass. Beyond this unified backward pass, the additional per-step computations introduced by EchoClip are also extremely lightweight: step-level entropy is computed via a simple reduction over the token-level logits that standard GRPO already produces, and EchoClip segments are identified by a single linear-time pass over the trajectory using these pre-computed entropies and a thresholding rule. Both operations are vectorized on the GPU and reuse the same rollout log-probabilities and rewards as vanilla GRPO, so the marginal cost of entropy calculation, segment identification, masking, summation, and advantage computation remains within profiler noise compared to the baseline. We provide a detailed algorithmic complexity analysis and visual runtime breakdown in Appendix F.

### 5.4. Framework Analysis (RQ3)

**Ablation Study.** We ablate the key design choices in `EchoRL` from the following perspectives: ① **The only new hyperparameter: loss weight** $\lambda$. Our method introduces a

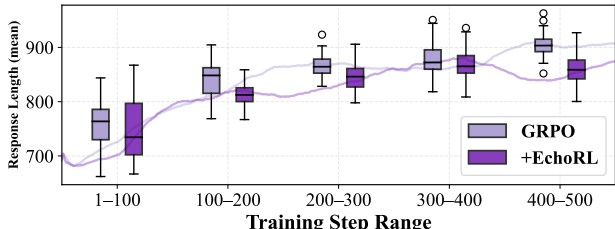

*Figure 7.* Mean response length during training for GRPO vs. GRPO+`EchoRL`. EchoRL tracks the same length trend as vanilla GRPO and does not introduce a systematic length inflation.

single additional hyperparameter $\lambda$ to weight the auxiliary loss. Following prior work (Li et al., 2025b; Liu et al., 2026; Li et al., 2026b), we default to $\lambda = 0.001$ to roughly align gradient magnitudes. As shown in Figure 6 (Left), larger values ($\lambda \geq 0.1$) degrade performance by overpowering the RL signal, whereas $\lambda = 0.001$ consistently improves stability. While fine-tuning $\lambda$ per model might yield marginal gains, we adhere to this robust default to demonstrate generalization. ② **Auxiliary target and training schedule.** Beyond $\lambda$, we ablate (i) *how* to form the auxiliary target from verified rollouts and (ii) *when* to apply EchoRL during training: EchoClip-based supervision consistently outperforms full-rollout and low-entropy variants, and late-stage activation can help (especially for DAPO) while GRPO prefers full training; to avoid introducing an extra schedule hyperparameter, we adopt the similarly strong full-training setup in the main paper (Appendix C). ③ **Length and stylistic effects.** A natural concern is that, because EchoRL supervises prefixes up to the most uncertain step, it might mainly encourage copying longer prefixes rather than better decision-making, leading to length-driven gains or stylistic overfitting. However, Figure 7 shows that the mean response length of GRPO+`EchoRL` closely tracks that of vanilla GRPO across training (and is sometimes slightly shorter), indicating that EchoRL does not simply "win by making answers longer" and that its gains are not explained by a trivial length preference.

**Sensitivity & Statistical Significance.** Addressing the concern that gains in RLVR can be difficult to ascertain without confidence intervals, we conducted rigorous sensitivity testing. For each baseline and `EchoRL`, we repeated the full evaluation pipeline across three independent runs on `Qwen2.5-Math-7B`. Figure 6 (Right) reports the mean performance with 95% confidence intervals (calculated as mean $\pm 1.96 \times \text{std}/\sqrt{3}$). The non-overlapping confidence bands confirm that `EchoRL` provides statistically significant improvements rather than random fluctuations and is not sensitive to randomness in individual runs.

## 6. Conclusion

`EchoRL` addresses the critical bottleneck of advantage degeneration in RLVR by extracting usable learning signals from high-entropy *EchoClip*s within verified rollouts. As a lightweight, plug-and-play module, it enables continuous learning even when reward variance collapses. Extensive experiments confirm that `EchoRL` consistently enhances reasoning performance across diverse models and tasks with negligible overhead, offering a robust and efficient solution for scaling up post-training.

## Impact Statement

This paper proposes a technical improvement to reinforcement learning post-training for large language models. The primary impact is to enhance training efficiency and model performance without introducing additional data or external supervision. As with all large language models, potential downstream applications may have broad societal implications; however, we do not believe the specific techniques introduced here introduce new risks beyond those already present in existing language model technologies.

## Acknowledgements

The authors gratefully acknowledge the scientific support and HPC resources provided by the Erlangen National High Performance Computing Center (NHR@FAU) of the Friedrich-Alexander-Universität Erlangen-Nürnberg (FAU) under the NHR project b270dd and b271dd. NHR funding is provided by federal and Bavarian state authorities. NHR@FAU hardware is partially funded by the German Research Foundation (DFG) – 440719683.

We acknowledge the EuroHPC Joint Undertaking for awarding this project access to the EuroHPC supercomputer LEONARDO, hosted by CINECA (Italy) and the LEONARDO consortium through an EuroHPC AI and Data-Intensive Applications Access call EHPC-AI-2024A06-060.

We further acknowledge financial support from BMFTR and SMWK within the Center of Excellence for AI Research "ScaDS.AI Dresden/Leipzig". We also acknowledge computing resources provided by the NHR Center of TU Dresden, jointly supported by BMFTR and the state governments participating in the NHR.

This work is partially supported by the NTU AI-for-X Postdoctoral Fellowship.

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

## A. Additional Results Across Backbones

This section reports additional results to assess the portability of `EchoRL` beyond our main `Qwen2.5-Math-7B` setting. Table 2 summarizes performance across three backbones (`Qwen2.5-Math-1.5B`, `Qwen2.5-7B-Instruct`, `Llama3.1-8B-Instruct`) on in-distribution (AIME24/AIME25/AMC/MATH-500/Minerva/Olympiad) and out-of-distribution (ARC-c/GPQA*/MMLU-Pro) benchmarks, reporting per-task scores and the corresponding ID/OOD averages.

*Table 2.* Overall in-distribution and out-of-distribution performance based on different backbone models. Bold indicates the best results within a comparable group.

| Model | In-Distribution Performance | | | | | | | Out-of-Distribution Performance | | | |
|---|---|---|---|---|---|---|---|---|---|---|---|
| | AIME24 | AIME25 | AMC | MATH-500 | Minerva | Olympiad | Avg. | ARC-c | GPQA* | MMLU-Pro | Avg. |
| *Qwen2.5-Math 1.5B* | | | | | | | | | | | |
| Base | 7.2 | 3.6 | 26.4 | 28.0 | 9.6 | 21.2 | 16.0 | 3.1 | 3.0 | 2.3 | 2.8 |
| GRPO | 10.5 | 6.2 | 38.5 | 63.5 | 25.5 | 34.0 | 29.7 | 54.0 | 26.5 | 28.5 | 36.3 |
| ↪ + EchoRL | 12.1 | 8.3 | 43.5 | 64.0 | 24.7 | 37.6 | **31.7** | 57.5 | 28.1 | 31.5 | **39.0** |
| *Qwen2.5-7B Instruct* | | | | | | | | | | | |
| Instruct | 11.7 | 7.5 | 43.8 | 71.8 | 30.9 | 40.4 | 34.4 | 84.7 | 24.7 | 54.7 | 54.7 |
| GRPO | 12.5 | 7.0 | 41.0 | 71.5 | 31.5 | 35.5 | 33.1 | 88.5 | 17.5 | 54.5 | 53.5 |
| ↪ + EchoRL | 14.8 | 7.7 | 41.8 | 76.9 | 35.4 | 42.5 | **36.5** | 90.2 | 21.2 | 59.5 | **56.9** |
| *Llama3.1-8B Instruct* | | | | | | | | | | | |
| Instruct | 4.0 | 0.3 | 15.7 | 39.8 | 18.0 | 13.3 | 15.2 | 42.4 | 0.0 | 40.5 | 27.6 |
| GRPO | 2.5 | 0.5 | 19.5 | 46.5 | 22.5 | 17.5 | 18.1 | 84.5 | 0.0 | 45.5 | 43.3 |
| ↪ + EchoRL | 7.2 | 0.8 | 24.2 | 49.8 | 25.8 | 20.7 | **21.4** | 88.1 | 11.4 | 49.2 | **49.6** |

## B. Sensitivity Analysis

*Table 3.* **Sensitivity Analysis on `Qwen2.5-Math-7B`.** We report the mean and 95% confidence interval (calculated as mean $\pm$ 1.96 $\times$ std/$\sqrt{3}$) across 3 independent runs. `EchoRL` consistently outperforms baselines while maintaining comparable stability.

| Method | In-Distribution Performance | | | | | | | Out-of-Distribution Performance | | | |
|---|---|---|---|---|---|---|---|---|---|---|---|
| | AIME24 | AIME25 | AMC | MATH-500 | Minerva | Olympiad | Avg. | ARC-c | GPQA* | MMLU-Pro | Avg. |
| GRPO | 26.1 ± 1.0 | 16.7 ± 0.4 | 60.5 ± 0.7 | 80.3 ± 1.4 | 40.0 ± 0.4 | 43.7 ± 1.6 | 44.6 ± 0.4 | 81.8 ± 1.1 | 39.3 ± 0.6 | 45.0 ± 0.4 | 55.4 ± 0.4 |
| ↪ + EchoRL | 24.6 ± 0.4 | 22.1 ± 0.8 | 62.4 ± 0.4 | 81.6 ± 0.6 | 40.9 ± 0.6 | 48.9 ± 1.0 | **46.8 ± 0.3** | 84.5 ± 0.8 | 37.4 ± 1.5 | 53.8 ± 0.8 | **58.6 ± 0.6** |
| DAPO | 29.3 ± 0.7 | 18.9 ± 0.4 | 62.8 ± 0.7 | 86.5 ± 1.5 | 40.3 ± 0.7 | 48.4 ± 0.4 | 47.7 ± 0.3 | 81.8 ± 0.7 | 39.6 ± 0.8 | 52.6 ± 0.8 | 58.0 ± 0.4 |
| ↪ + EchoRL | 33.9 ± 0.7 | 21.9 ± 0.7 | 62.3 ± 0.8 | 88.1 ± 0.4 | 41.0 ± 0.8 | 50.2 ± 0.5 | **49.6 ± 0.3** | 85.1 ± 1.1 | 43.0 ± 1.5 | 54.7 ± 1.0 | **61.1 ± 0.7** |

To further validate the robustness of `EchoRL`, we conducted a sensitivity analysis examining the performance variance across multiple runs. For GRPO and DAPO baselines, as well as their `EchoRL`-augmented counterparts, we repeated the evaluation under the same experimental setup, reporting the performance mean and 95% confidence interval (calculated as $1.95 \times$ std/$\sqrt{3}$) across runs to average out randomness in individual trials.

As shown in Table 3, `EchoRL` not only improves the mean performance across almost all benchmarks but also maintains stable variance. The confidence intervals indicate that the improvements are consistent and not due to random fluctuations.

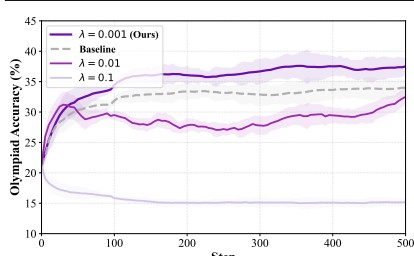

| Method | ID Avg. | OOD Avg. |
|---|---|---|
| GRPO | 44.5 | 55.6 |
| ↪ Full High-Entropy Rollout | 45.2 | 56.1 |
| ↪ Lowest-Entropy Rollout | 42.9 | 54.0 |
| ↪ + EchoRL (Full) | **47.0** | **58.4** |
| DAPO | 47.9 | 57.6 |
| ↪ Full High-Entropy Rollout | 48.3 | 58.0 |
| ↪ Lowest-Entropy Rollout | 46.1 | 56.2 |
| ↪ + EchoRL (Full) | **49.9** | **60.9** |

| Method | ID Avg. | OOD Avg. |
|---|---|---|
| GRPO | 44.5 | 55.6 |
| ↪ + EchoRL (1–200) | 44.6 | 55.7 |
| ↪ + EchoRL (300–500) | 46.8 | 58.2 |
| ↪ + EchoRL (Full) | **47.0** | **58.4** |
| DAPO | 47.9 | 57.6 |
| ↪ + EchoRL (1–200) | 49.6 | 60.6 |
| ↪ + EchoRL (300–500) | **50.3** | **61.3** |
| ↪ + EchoRL (Full) | 49.9 | 60.9 |

*Figure 8.* **Detailed Ablation Study (Qwen2.5-Math-7B). Left:** Sweep of the auxiliary-loss weight $\lambda$ (larger values can overpower the RLVR signal; $\lambda = 0.001$ yields the best overall performance, while $\lambda = 0.1$ degrades). **Middle:** Entropy/clip-selection ablation (Avg-only, ID/OOD Avg) comparing full verified rollout vs lowest-entropy rollout clip vs EchoRL trained throughout. **Right:** EchoRL activation window ablation (Avg-only). The step range is shown in parentheses after the arrow: for GRPO, full training is best; for DAPO, a later window (300–500) can be slightly better than full, but introduces an extra schedule hyperparameter, so we use the simpler full-training mechanism in the main paper.

## C. Detailed Ablation Study

**What is ablated?** We isolate two core design choices while keeping all other training settings fixed (same data, seeds, rollout budget, and optimizer hyperparameters).

**Auxiliary-loss weight $\lambda$.** The left panel of Figure 8 sweeps $\lambda$ and shows a clear sweet spot: $\lambda = 0.001$ consistently achieves the best average performance. Larger weights (e.g., $\lambda = 0.1$) can over-emphasize the auxiliary objective relative to the RLVR signal and noticeably degrade final performance.

**Clip/step selection strategy.** The right table in Figure 8 compares three concrete ways to define the auxiliary supervision target from the verified-success set $\mathcal{V}$:

- Full High-Entropy Rollout: apply the auxiliary loss to the entire verified rollout tokens (i.e., no truncation; the auxiliary target is the full successful trajectory).

- Lowest-Entropy Rollout: apply the same auxiliary loss to a low-uncertainty successful trajectory clip (selected as the verified rollout with the lowest average step entropy).

- EchoClip: select the single highest-entropy step across all verified rollouts, then take the prefix of the rollout containing that step, ending exactly at that step; the auxiliary loss is applied only on this prefix.

Across both GRPO and DAPO, EchoClip yields the strongest ID/OOD gains, while the lowest-entropy variant underperforms the baseline, consistent with the intuition that focusing supervision on the pivotal uncertain decision point is crucial.

**When to enable EchoRL during training.** We further ablate the timing of applying the EchoRL auxiliary loss (fixing $\lambda = 0.001$ and using the EchoClip construction) by enabling it only in early training (steps 1–200), only in mid/late training (steps 300–500), or throughout the entire run (summarized in the right table of Figure 8). For GRPO, full training performs best, while enabling EchoRL only in the first 200 steps is essentially on par with the GRPO baseline. For DAPO, enabling EchoRL only in steps 300–500 is slightly better than applying it throughout training. These trends align with the intuition that later training contains more high-quality verified-success rollouts (hence more informative EchoClips), whereas early-stage rollouts are noisier. However, windowed schedules introduce an additional step-range hyperparameter; in the main paper we therefore adopt the similarly strong and simpler full-training mechanism.

## D. Newline Tokens for Step Segmentation

Accurate step segmentation is critical for computing step-wise entropy in Chain-of-Thought reasoning. In mainstream BPE tokenizers, newline characters (\n) are typically encoded as specific Token IDs. After investigation and verification, we uniformly configured `newline_token_ids = [198, 271]` across all experiments, covering our main `Qwen2.5-Math-7B` setting and the additional backbones in Appendix A (`Qwen2.5-Math-1.5B`, `Qwen2.5-7B Instruct`, `Llama3.1-8B Instruct`), as well as `Qwen2.5-VL-7B`.

**Token ID Analysis.**

- **ID 198 (Ġ / \n):** The most universal newline token. In both `LLaMA-3` (Tiktoken-based) and `Qwen2.5` (Qwen-Tiktoken-based) vocabularies, ID 198 corresponds to the standard newline character. It identifies the natural line breaks in the vast majority of reasoning steps.

- **ID 271 (ĠĠ / \n\n):** Primarily for the `Qwen2.5` family. In `Qwen2.5` tokenizers, 271 often corresponds to double newlines or specific whitespace combinations. Including this ID improves segmentation robustness against non-standard formatting (e.g., Markdown lists) or large paragraph breaks.

**Compatibility Note.**  Although `LLaMA-3.1` primarily uses 198, including 271 is safe. If the model does not generate ID 271, the algorithm simply treats it as an unused delimiter without affecting the standard segmentation via ID 198.

*Table 4.* Newline Token Configuration across Models

| Model | Tokenizer Type | Key Newline ID | Configured IDs |
|---|---|---|---|
| Llama3.1-8B Instruct | Tiktoken (gpt-4 based) | 198 | [198, 271] |
| Qwen2.5-Math-7B | Qwen-Tiktoken | 198, 271 | [198, 271] |
| Qwen2.5-Math-1.5B | Qwen-Tiktoken | 198, 271 | [198, 271] |
| Qwen2.5-7B Instruct | Qwen-Tiktoken | 198, 271 | [198, 271] |
| Qwen2.5-VL-7B | Qwen-Tiktoken | 198, 271 | [198, 271] |

# E. EchoRL

EchoRL is applied to prompts with at least one successful rollout. If $\mathcal{V}$ is empty, skip the EchoRL auxiliary loss for this prompt.

---
**Algorithm 1 EchoRL**

---
1: **Input:** $\mathcal{D}, \pi_\theta, \lambda$. **Output:** $\pi_\theta$.
2: **for** each training step **do**
3:   **Sample & Verify:** Sample $q \sim \mathcal{D}$, generate $\{o^{(i)}\}$, compute rewards.
4:   $\mathcal{V} \leftarrow \{o^{(i)} \mid r(o^{(i)}) = 1\}$ {▷ Filter successful rollouts}
5:   **EchoClip Mining:**
6:   Extract all steps: $\mathcal{S} \leftarrow \bigcup_{o \in \mathcal{V}} \text{Segment}(o)$
7:   Find hardest step: $s^\star \leftarrow \operatorname{argmax}_{s \in \mathcal{S}} \bar{H}(s)$
8:   Extract Clip: $o_{\text{echo}} \leftarrow \text{Prefix}(o(s^\star), s^\star)$ {▷ Clip ending at max-entropy step}
9:   **EchoRL Update:**
10:   $J_{\text{EchoRL}} \leftarrow -\frac{1}{|o_{\text{echo}}|} \sum \log \pi_\theta(o_{\text{echo}})$
11:   $\theta \leftarrow \theta - \eta \nabla(J_{\text{RLVR}} + \lambda J_{\text{EchoRL}})$ {▷ Update with auxiliary loss}
12: **end for**

---

# F. Algorithmic Complexity and Computational Overhead

To substantiate the "free lunch" efficiency claim, we provide a formal complexity analysis of `EchoRL` compared to standard GRPO.

**Time Complexity Decomposition.** Let $T$ denote the average rollout length and $N$ the number of rollouts per prompt. The training loop consists of a generation phase (rollout) and an update phase. In the update phase, standard GRPO computes the policy gradient loss $\mathcal{L}_{\text{GRPO}}$ and performs backpropagation. `EchoRL` modifies this process by introducing an auxiliary loss term $\mathcal{L}_{\text{EchoRL}}$ computed on *EchoClip*s—subsequences of the verified-success rollouts. Crucially, `EchoRL` does **not** require an additional backward pass. Instead, we effectively merge the objectives into a single scalar $\mathcal{L}_{\text{total}} = \mathcal{L}_{\text{GRPO}} + \lambda \mathcal{L}_{\text{EchoRL}}$ and execute a **unified backward pass**. Since the logits required for $\mathcal{L}_{\text{EchoRL}}$ are a subset of those already computed for $\mathcal{L}_{\text{GRPO}}$

(obtained via the same forward pass over the rollout), the auxiliary term essentially involves applying a specific mask to the existing computation graph. Consequently, the additional arithmetic operations (masking and summation) have $O(N \cdot T)$ complexity, which is negligible compared to the $O(N \cdot T \cdot d_{\mathrm{model}}^2)$ complexity of the transformer's gradient computation.

**Latency Analysis.** We empirically evaluate the real-world overhead by monitoring the "Actor Update Time" (the duration of the optimization step) throughout the training process. As illustrated in Figure 9, the update latency for `EchoRL` closely tracks that of the GRPO baseline, with both methods exhibiting similar fluctuations in the range of 40–55 seconds. The overlapping curves indicate that the cost of the unified backward pass in `EchoRL` is statistically indistinguishable from the baseline. This validates our complexity analysis: by reusing the computation graph and merging gradients, `EchoRL` enhances data efficiency without incurring the latency penalties typically associated with auxiliary supervision or dual-objective methods.

**Memory Footprint.** `EchoRL` operates entirely within the VRAM allocated for GRPO. The auxiliary loss is computed by reusing the policy model's computation graph. No external reward models, critic networks, or reference models are loaded into memory beyond what is required by the base RLVR algorithm. The peak memory usage remains determined by the activation memory required for the longest rollout in the batch, which is unchanged by our method.

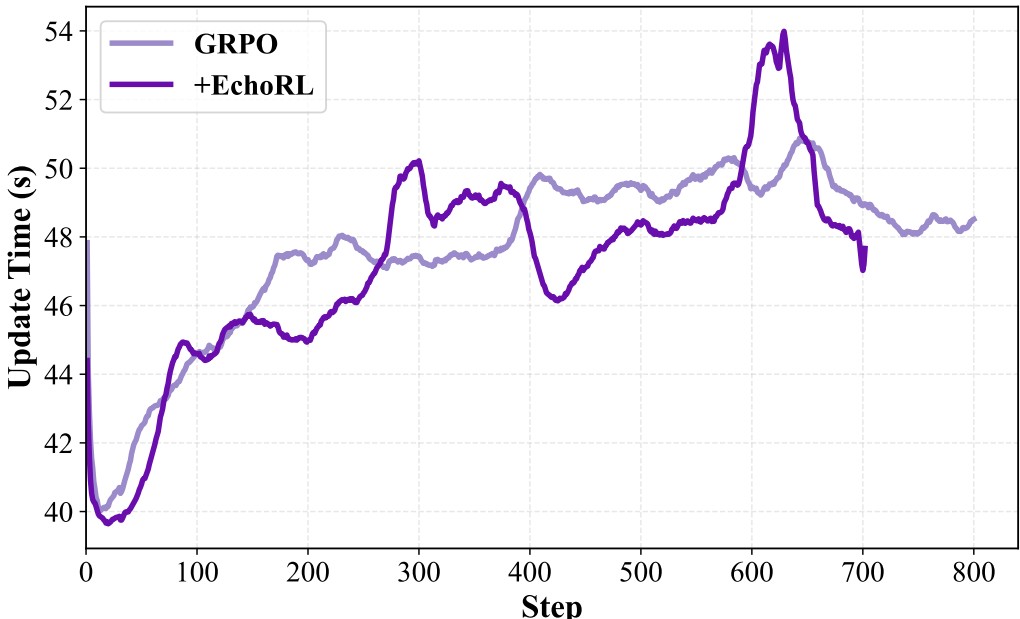

*Figure 9.* **Actor Update Latency.** The update time (in seconds) for `EchoRL` and GRPO across training steps. The intertwined curves demonstrate that `EchoRL` introduces no significant computational overhead to the update phase.

## G. Training Details

This appendix summarizes the training and evaluation settings used throughout our experiments. Table 5 reports the key RLVR hyperparameters (data, actor, rollout, and trainer configurations), while the evaluation protocol below specifies our decoding and scoring conventions.

**Evaluation protocol.** For AIME and AMC benchmarks with limited numbers of samples, we report Avg@32 over 32 independent runs; for other benchmarks we report Pass@1 (the solved proportion under a single attempt). All evaluations use sampling temperature 0.6 and top-$p$ 1.0, with answers extracted and verified via Math-Verify.[3]

---

[3] https://github.com/huggingface/Math-Verify

*Table 5.* RLVR training hyperparameters used in our experiments. All sequence lengths are measured using each model's own tokenizer.

| Module | Parameter | Value | Description |
|---|---|---|---|
| Data | `data.train_batch_size` | 128 | Global training batch size per optimization step. |
| | `data.val_batch_size` | 512 | Batch size used for validation/evaluation. |
| | `data.max_prompt_length` | 1024 | Maximum input length. |
| | `data.max_response_length` | 8192 | Maximum response length. |
| Actor | `actor_rollout_ref.actor.optim.lr` | $1 \times 10^{-6}$ | Learning rate for the actor optimizer. |
| | `actor_rollout_ref.actor.ppo_mini_batch_size` | 64 | PPO mini-batch size. |
| | `actor_rollout_ref.actor.ppo_micro_batch_size` | 64 | PPO micro-batch size. |
| | `actor_rollout_ref.actor.entropy_coeff` | 0.001 | Entropy coefficient in the RLVR objective. |
| Rollout | `actor_rollout_ref.rollout.engine` | vllm | Inference/rollout engine. |
| | `actor_rollout_ref.rollout.temperature` | 1.0 | Sampling temperature during training rollouts. |
| | `actor_rollout_ref.rollout.val_temperature` | 0.6 | Sampling temperature during evaluation. |
| | `actor_rollout_ref.rollout.gpu_memory_utilization` | 0.80 | Fraction of GPU memory to utilize for the rollout engine. |
| | `actor_rollout_ref.rollout.n` | 8 | Number of rollouts per prompt during RLVR. |
| Trainer | `trainer.critic-warmup` | 0 | Number of warmup steps (0 disables warmup). |
| | `trainer.training_steps` | 700/500 | Number of training steps (700 for `Qwen2.5-Math-7B`, 500 for other models). |

## G.1. Evaluation Benchmarks

We evaluate models on eight benchmarks grouped into mathematical reasoning (in-distribution) and out-of-distribution (OOD) general reasoning tasks. Table 6 summarizes dataset sizes and metadata.

### G.1.1. MATHEMATICAL REASONING BENCHMARKS

**AIME24/AIME25.** Problems from the American Invitational Mathematics Examination (AIME)[4] with integer answers in $[0, 999]$, covering algebra, geometry, trigonometry, number theory, probability, and combinatorics. We use the 2024 and 2025 editions as separate test sets.

**AMC.** Competition problems from AMC12 (2022–2023), extracted from the AoPS wiki[5]; this set serves as an internal validation benchmark for competition-style math reasoning.

**MATH-500.** A 500-problem subset sampled from the MATH dataset (Hendrycks et al., 2021), used as a clean evaluation set for multi-step mathematical reasoning.

**Minerva.** A mathematical reasoning benchmark spanning high-school to undergraduate topics and requiring multi-step symbolic reasoning (Lewkowycz et al., 2022).

**OlympiadBench.** Olympiad-level mathematics problems with fine-grained domain categorization; we use the OE_TO_MATHS_EN_COMP subset (He et al., 2024).

---

[4] https://maa.org/math-competitions/aime
[5] https://artofproblemsolving.com/wiki/index.php/Main_Page

*Table 6.* Benchmarks used in this study. "–" indicates the split is not officially provided.

| Dataset | #Train | #Test | Task Type | Domain | License | Source |
|---------|--------|-------|-----------|--------|---------|--------|
| *Mathematical Reasoning Benchmarks* | | | | | | |
| AIME24 | – | 30 | Math competition | Mathematics | MIT | AIME |
| AIME25 | – | 30 | Math competition | Mathematics | MIT | AIME |
| AMC | – | 83 | Math competition | Mathematics | Apache 2.0 | AoPS Wiki |
| MATH-500 | – | 500 | Mathematical reasoning | Mathematics | – | (Hendrycks et al., 2021) |
| Minerva | – | 272 | Mathematical reasoning | Mathematics | Apache 2.0 | (Lewkowycz et al., 2022) |
| OlympiadBench | – | 674 | Math competition | Mathematics | Apache 2.0 | (He et al., 2024) |
| *Out-of-Distribution (OOD) Benchmarks* | | | | | | |
| ARC-c | – | 1,172 | Science reasoning | General science | CC-BY-SA-4.0 | (Clark et al., 2018) |
| GPQA-Diamond | – | 198 | Scientific reasoning | Bio/Phys/Chem | CC-BY-4.0 | (Rein et al., 2024) |
| MMLU-Pro | – | 12,032 | Multi-task understanding | Multidisciplinary | MIT | (Wang et al., 2024a) |

G.1.2. OUT-OF-DISTRIBUTION (OOD) BENCHMARKS

**ARC-c.** Grade-school science questions requiring commonsense reasoning and knowledge application (Clark et al., 2018).

**GPQA-Diamond.** Graduate-level multiple-choice questions in biology/physics/chemistry; we use the Diamond split (Rein et al., 2024).

**MMLU-Pro.** An enhanced, more challenging version of MMLU for broad multi-task language understanding across disciplines (Wang et al., 2024a).

# H. Case Study: Usable Signals under Advantage Degeneration

This appendix expands the example shown in Figure 1. We present the full prompt and four verified-success rollouts (all with identical verifiable reward), and answer four questions in a step-by-step manner to clarify why standard group-relative objectives can fail to reinforce the best reasoning path.

**Q1: What is the prompt and what is the evaluation setting?**

**A1:** To set up the case study, we need to pin down (i) the prompt and (ii) what "advantage-degenerated" means here.

1. **Prompt**: the quartic-polynomial question shown below.

2. **Setting**: all rollouts are sampled from the same checkpoint and all pass verification (reward $r = 1$).

---
**Prompt (Quartic Polynomial / AIME-style)**

```
7. Given a quartic polynomial $f(x)$ whose four real roots form an arithmetic sequence
    with a common difference of 2. Then the difference between the largest and
    smallest roots of $f^{\prime}(x)$ is $\qquad$
```
---

**Q2: What are the four rollouts (R1–R4)?**

**A2:** We list the full trajectories (verbatim) below. Even though all four rollouts reach the correct answer, they follow meaningfully different solution paths:

1. **R1–R3**: use a routine algebraic route (center/scale the roots, expand into a polynomial, then differentiate and factor).

2. **R4**: uses the log-derivative identity for $g(x) = \prod_i (x - r_i)$, where critical points satisfy $\sum_i \frac{1}{x-r_i} = 0$, yielding the same critical points with less algebra.

---

**R1 (Differentiate + factor after centering)**

```
Thought
<think>
To solve the problem, we need to find the difference between the largest and smallest
    roots of the derivative of the quartic polynomial.
Here is the step-by-step thought process:
1. \textbf{Understand the Problem}: We are given a quartic polynomial $f(x)$ with four
    real roots in an arithmetic sequence with a common difference of 2. We need to
    find the difference between the largest and smallest roots of the derivative $f^{\
    prime}(x)$.
2. \textbf{Use Symmetry}: The roots are evenly spaced. We can shift the polynomial to
    center the roots at the origin. This simplifies calculations because the polynomial
    becomes an even function, and the spacing between critical points is invariant
    under translation.
3. \textbf{Define the Shifted Polynomial}: Let the centered roots be $-3, -1, 1, 3$ (
    since the common difference is 2). The polynomial in the shifted variable $y$ is $g
    (y) = (y-3)(y-1)(y+1)(y+3)$.
4. \textbf{Differentiate}: Expand $g(y)$ into a polynomial form and compute its
    derivative $g^{\prime}(y)$.
5. \textbf{Find the Roots of the Derivative}: Set $g^{\prime}(y) = 0$ and solve for
    $y$ to find the critical points.
6. \textbf{Calculate the Difference}: Identify the largest and smallest roots of $g^{\
    prime}(y)$ and calculate their difference. This will be the final answer.
Let's use these steps to solve the problem.
</think>
Solution
Let the roots be $a,a+2,a+4,a+6$ and set $m=a+3$. Put $y=x-m$, so the roots become $
    -3,-1,1,3$. Then
\[g(y)=(y-3)(y-1)(y+1)(y+3)=(y^2-9)(y^2-1)=y^4-10y^2+9.\]
Differentiate:
\[g'(y)=4y^3-20y=4y(y^2-5).\]
Hence the critical points are at $y=0,\pm\sqrt5$. Translating back does not change the
     difference between largest and smallest roots, so
\[\Delta = \sqrt5-(-\sqrt5)=2\sqrt5.\]
Answer: \boxed{2\sqrt5}.
```

---

**R2 (Shift + scale, then differentiate)**

```
Thought
<think>
To find the difference between the largest and smallest roots of $f^{\prime}(x)$, we
    can use a linear transformation to simplify the arithmetic.
Here is the step-by-step thought process:
1. \textbf{Understand the Goal}: We need to find the range of the roots of the
    derivative of a quartic polynomial whose roots form an arithmetic progression with
    difference 2.
2. \textbf{Apply Transformation}: We can shift and scale the roots to make the numbers
     smaller and symmetric. Let's map the roots to $\pm \frac{1}{2}, \pm \frac{3}{2}$.
    This corresponds to a scaling factor of $1/2$ (since the original difference is 2
    and the new difference is 1).
3. \textbf{Form Polynomial in Scaled Variable}: Construct the polynomial $G(z)$ with
    these simpler roots.
4. \textbf{Differentiate and Solve}: Find the roots of the derivative $G^{\prime}(z)$.
5. \textbf{Rescale and Conclude}: Convert the roots of $G^{\prime}(z)$ back to the
    original scale. Since we scaled the coordinates by $1/2$, the distances in the $z$-
    domain are half the distances in the $x$-domain. We must multiply the result by 2
    to get the final answer.
Let's apply this transformation and solve.
</think>
```

```
Solution
Roots: $a,a+2,a+4,a+6$. Let $m=a+3$ and define
\[z=\frac{x-m}{2}\quad (x=m+2z).\]
Then the roots in $z$ are $-\tfrac32,-\tfrac12,\tfrac12,\tfrac32$. Up to a nonzero
    constant,
\[G(z)=(z^2-\tfrac14)(z^2-\tfrac94)=z^4-\tfrac52 z^2+\tfrac{9}{16}.\]
Differentiate:
\[G'(z)=4z^3-5z=z(4z^2-5).\]
So $z=0,\pm\tfrac{\sqrt5}{2}$. Converting back to $x$ gives
\[x=m+2z\in\{m,\ m\pm\sqrt5\}.\]
Thus the largest-minus-smallest root of $f'(x)$ is
\[(m+\sqrt5)-(m-\sqrt5)=2\sqrt5.\]
Therefore \boxed{2\sqrt5}.
```

## R3 (Parity argument + product rule)

```
Thought
<think>
To solve this problem efficiently, we can exploit the symmetry and parity properties
    of the polynomial.
Here is the step-by-step thought process:
1. \textbf{Analyze the Roots}: The roots are in an arithmetic progression. Centering
    them at 0 yields the set $\{-3, -1, 1, 3\}$.
2. \textbf{Identify Parity}: The centered polynomial $g(y)$ has roots symmetric around
     0, which means $g(y)$ is an even function ($g(y) = g(-y)$).
3. \textbf{Derivative Properties}: The derivative of an even function is an odd
    function. This immediately implies that $g^{\prime}(0) = 0$, so $y=0$ is one
    critical point.
4. \textbf{Find Other Roots}: Differentiate $g(y)$ (using the product rule on grouped
    factors like $(y^2-1)(y^2-9)$) to find the remaining non-zero roots.
5. \textbf{Calculate Difference}: The non-zero roots will be symmetric, say $\pm c$.
    The difference between the largest and smallest roots is $c - (-c) = 2c$.
Let's use this parity argument to find the answer.
</think>
Solution
Center the roots: let $m=a+3$ and $y=x-m$, so the roots are $-3,-1,1,3$. Then
\[g(y)=(y^2-1)(y^2-9).\]
Differentiate using the product rule:
\[g'(y)=2y(y^2-9)+2y(y^2-1)=4y^3-20y=4y(y^2-5).\]
So $g'(y)=0$ at $y=0,\pm\sqrt5$. Translating back does not change the spread, hence
    the difference between the largest and smallest roots of $f'(x)$ is
\[\sqrt5-(-\sqrt5)=2\sqrt5.\]
Answer: \boxed{2\sqrt5}.
```

## R4 (Log-derivative trick)

```
Thought
<think>
To find the critical points without full expansion, we can use the logarithmic
    derivative property.
Here is the step-by-step thought process:
1. \textbf{Center the Roots}: Shift the polynomial so the roots become symmetric
    around zero: $\{-3, -1, 1, 3\}$. Let the shifted variable be $y$.
2. \textbf{Use Logarithmic Derivative}: For a polynomial $g(y) = \prod (y-r_i)$, the
    critical points satisfy $\frac{g^{\prime}(y)}{g(y)} = \sum \frac{1}{y-r_i} = 0$.
3. \textbf{Set up the Equation}: Substitute the roots into the sum: $\frac{1}{y-3} + \
    frac{1}{y-1} + \frac{1}{y+1} + \frac{1}{y+3} = 0$.
4. \textbf{Group Terms}: Pair the terms with opposite roots (e.g., $\frac{1}{y-3} + \
```

```
         frac{1}{y+3}$) to simplify the algebra using difference of squares.
   5. \textbf{Solve for y}: Solve the resulting rational equation to find the values of
      $y$ where the derivative is zero.
   6. \textbf{Compute Difference}: Determine the largest and smallest roots from the
      solution set and compute their difference.
   Let's solve using this method.
   </think>
   Solution
   Let the roots be $a,a+2,a+4,a+6$ and set the midpoint $m=a+3$. Define $y=x-m$. Then
      the roots become $-3,-1,1,3$, so
   \[g(y)=(y+3)(y+1)(y-1)(y-3).\]
   For $g(y)\neq 0$, we have
   \[\frac{g'(y)}{g(y)}=\sum_{r\in\{-3,-1,1,3\}}\frac{1}{y-r},\]
   so $g'(y)=0$ is equivalent to
   \[\frac1{y+3}+\frac1{y+1}+\frac1{y-1}+\frac1{y-3}=0.\]
   Pair terms:
   \[\left(\frac1{y+3}+\frac1{y-3}\right)+\left(\frac1{y+1}+\frac1{y-1}\right)=\frac{2y}{
      y^2-9}+\frac{2y}{y^2-1}=0.\]
   Thus either $y=0$ or
   \[\frac1{y^2-9}+\frac1{y^2-1}=0\ \Rightarrow\ (y^2-1)+(y^2-9)=0\ \Rightarrow\ 2y
      ^2-10=0\ \Rightarrow\ y=\pm\sqrt5.\]
   So the roots of $f'(x)$ are $m-\sqrt5,\ m,\ m+\sqrt5$, and the requested difference is
   \[(m+\sqrt5)-(m-\sqrt5)=2\sqrt5.\]
   Therefore the answer is \boxed{2\sqrt5}.
```

## Q3: Why does GRPO/DAPO fail to reinforce the best reasoning path in this group?

**A3:** Although the trajectories differ qualitatively, they all receive the same verifiable reward ($r = 1$). The key issue is that GRPO/DAPO rely on within-group reward standardization:

1. **Identical rewards**: $r(R1) = \cdots = r(R4) = 1$ implies the group reward standard deviation is (near) zero.

2. **Standardization collapses**: with (near-)zero standard deviation, the standardized group-relative advantages satisfy $A(R1) = \cdots = A(R4) \approx 0$.

3. **Vanishing gradient**: the resulting policy-gradient contribution from this prompt is near zero, so the optimizer cannot prefer the higher-quality reasoning path (Figure 1).

## Q4: What signal does EchoRL extract here?

**A4:** EchoRL creates a usable learning signal by focusing supervision on the uncertain, high-entropy part of verified-success rollouts:

1. **Measure step entropy** within verified-success rollouts.

2. **Select the highest-entropy step** and take the corresponding prefix (EchoClip).

3. **Apply an auxiliary loss** on this EchoClip so that learning remains active even when group-relative advantages degenerate.

In this example, the rollout using the log-derivative identity contains a short, high-leverage reasoning step that is easy to miss under reward-only standardization but can be reinforced via EchoClip-based supervision.

# I. Prompt Template

## I.1. RLVR Training and Evaluation

We use the same prompt template for most models. However, due to the limited capability of the `Llama-3.1 8B` model, we adopt a simplified prompt to ensure stable zero-shot generation. For `Qwen2.5-VL`, we follow the default EasyR1 template used in our multimodal experiments.

## RLVR / Evaluation Prompt

```
Your task is to follow a systematic, thorough reasoning process before providing the
    final solution.
This involves analyzing, summarizing, exploring, reassessing, and refining your
    thought process
through multiple iterations. Structure your response into two sections: Thought and
    Solution.
In the Thought section, present your reasoning using the format: "<think>\n thoughts
</think>\n". Each thought should include detailed analysis, brainstorming,
    verification, and
refinement of ideas. After "</think>\n" in the Solution section, provide the final,
    logical,
and accurate answer, clearly derived from the exploration in the Thought section. If
    applicable,
include the answer in \boxed{} for closed-form results like multiple choices or
    mathematical
solutions.

User: This is the problem: {Question}
Assistant: <think>
```

## RLVR / Evaluation Prompt (Llama-3.1 8B Instruct)

```
User: {Question}
Answer: Let's think step by step.\n
```

## RLVR / Evaluation Prompt (EasyR1, Qwen2.5-VL)

```
{{ content | trim }} You FIRST think about the reasoning process as an internal
    monologue and then provide the final answer. The reasoning process MUST BE enclosed
     within <think> </think> tags. The final answer MUST BE put in \boxed{}.
```

