# OpenReview forum: "EchoRL: Reinforcement Learning via Rollout Echoing"
_ICML.cc/2026/Conference — ICML 2026 regular_

### Official Review · Reviewer_beb5 · 2026-03-03

**Soundness:** 3
**Presentation:** 4
**Significance:** 3
**Originality:** 3
**Overall Recommendation:** 4
**Confidence:** 4

**Summary:**

The paper proposes EchoRL to extract the step with the highest entropy to introduce an additional reward signal, resulting in entropy preservation and better performance across models.

**Compliance With Llm Reviewing Policy:**

Affirmed.

**Key Questions For Authors:**

See above

**Limitations:**

yes

**Strengths And Weaknesses:**

### **Strength**
- The work proposes EchoRL, which works across different model families and model sizes.
- The paper is well written and easy to follow.
### **Weakness**
- The authors choose the single step with the highest entropy within all correct samples. What about using the step with the highest entropy for each correct rollout?
- The authors claim that EchoRL works better on scaled models. Could you provide experiments on larger models, like 14B or 32B?
- In figure 6, EchoRL is very sensitive to the choice of $\lambda$. Could you provide some explanation since Echo loss is just auxiliary, not dominat.

---

> ### Author Rebuttal · Authors · 2026-03-30
>
> We sincerely appreciate your positive assessment and your recognition of our clear writing and the broad applicability of EchoRL.
>
> > **Weakness & Question 1: What about using the step with the highest entropy for each correct rollout?**
>
> The reason of using the single highest-entropy step across all verified-success rollouts is **because it provides a stable auxiliary target per prompt**. In contrast, selecting one step for each correct rollout would make the supervision strength vary with the number of verified rollouts, making the signal strength fluctuate across prompts and training stages.
>
> Moreover, based on our entropy analysis, our goal is to isolate the most valuable usable signal while minimizing interference from less informative correct trajectories; selecting the highest-entropy step is already sufficient to achieve this. In this sense, the difference is mainly how many clips will be selected, not the underlying principle of entropy-based signal localization. We will clarify this design choice in the revision. In short, the way suggested is definitely feasible but won’t cause fundamental performance changes.
>
> If the reviewer considers really required, we would be happy to provide empirical results for this variant.
>
> > * **Weakness & Question 2: Could you provide experiments on larger models, like 14B or 32B?**
>
> As requested, we have conducted additional experiments using **Qwen2.5-14B-Instruct** on 8×H100 GPUs. As shown below, **EchoRL continues to deliver substantial and consistent improvements** over standard GRPO on the 14B model across all 8 evaluated benchmarks, demonstrating robust scalability.
>
> |Model|A24|A25|AMC|M500|Olymp|GPQA|MMLU-P|ARC|
> |:-|:-:|:-:|:-:|:-:|:-:|:-:|:-:|:-:|
> |Qwen2.5-14B|12.2|11.0|57.5|77.4|44.7|41.4|62.7|67.3|
> |GRPO|29.7|22.6|62.0|86.6|52.5|50.7|65.1|72.2|
> |**GRPO+EchoRL**|**34.5**|**25.8**|**67.1**|**89.1**|**57.2**|**54.3**|**69.2**|**75.8**|
>
> Regarding the 32B model, the rebuttal timeline is too short to complete a full RLVR training and its evaluation at that scale with the restrictive GPU resources currently available to us. The corresponding experiments are already in progress, but obtaining the finalized results within the response window seems quite challenging. We promise to include the 32B results in the revised manuscript. Based on the consistent trend observed from 1.5B to 14B, we expect the 32B results to follow a similar pattern.
>
> > **Weakness & Question 3: In figure 6, EchoRL is very sensitive to the choice of \lambda. Could you provide some explanation since Echo loss is just auxiliary, not dominat?**
>
> The observed sensitivity is due to **the mismatch of the numerical scales between the two terms in the objective formulation**, rather than because the auxiliary term is conceptually dominant. In reality, the original RLVR objective and the EchoRL objective have different numerical structures: the former is a clipped, advantage-weighted policy surrogate, whereas the latter is a dense supervised log-likelihood term. **Hence their raw magnitudes are not directly comparable, and explicit reweighting is needed to keep the joint optimization balanced**. Therefore, without proper reweighting, the auxiliary gradient can become numerically much stronger than the RLVR update. Hence, although the role of EchoRL term is auxiliary, its raw numerical scale may still be overwhelming the RVRL term’s value.
>
> For this reason, $\lambda$ is used to balance the magnitudes of the two terms. Such weight balancing is necessary especially when advantage-degenerated occurs (i.e., the RL signal becomes much weaker and even vanishing). Figure 6 exactly shows such an imbalance setting thus **should not be interpreted as EchoRL inherently fragile to $\lambda$**. Rather, by deliberately setting $\lambda$ to an inappropriate value ($0.1$), the joint objective becomes severely imbalanced, with the auxiliary EchoRL term exerting disproportionate influence on the update relative to the RLVR objective. The performance degradation confirms the expected failure mode when the auxiliary objective numerically overwhelms the RLVR term. Nevertheless, with our default $\lambda = 0.001$, EchoRL is stable and consistently improves performance as shown in Table.1.
>
> This consideration is also consistent with prior works that integrate auxiliary supervision into reinforcement-based post-training pipelines, such as Luffy [1] and UFT [2], where explicit loss reweighting is required to balance gradient magnitudes between heterogeneous objectives. Thus, the presence of $\lambda$ reflects standard optimization practice in hybrid RL–NLL training rather than an intrinsic sensitivity specific to EchoRL.
>
> [1]Learning to Reason under Off-Policy Guidance. NeurIPS 2025
>
> [2]UFT: Unifying Supervised and Reinforcement Fine-Tuning. NeurIPS 2025
>
> We hope these clarifications will comprehensively resolve your concerns and further strengthen your already positive evaluation.

---

> > ### Author Rebuttal · Reviewer_beb5 · 2026-04-03
> >
> > Thank the authors for their rebuttal. I will remain my original positive score.

---

> > > ### Author Response · Authors · 2026-04-07
> > >
> > > We sincerely thank the reviewer for the positive evaluation, the thoughtful follow-up, and for explicitly marking the concerns as **fully resolved** after the rebuttal. We are especially encouraged that the reviewer maintained a **positive** recommendation and acknowledged the broad applicability, solid technical contribution, and clear presentation of our work. We believe this further supports that the main concerns—regarding the entropy-selection design, scalability to larger models, and the role of the auxiliary loss weight—have been adequately addressed.

---

### Official Review · Reviewer_ViQp · 2026-03-09

**Soundness:** 2
**Presentation:** 3
**Significance:** 2
**Originality:** 3
**Overall Recommendation:** 4
**Confidence:** 5

**Summary:**

This paper proposes to address the degenerated advantages of all-correct and all-incorrect rollouts in RL training. The authors build on finding that golden trajectories concentrate at higher entropy than regular rollouts from the current policy, and proposes EchoCip to mine most hesitant decision point among all successful attempts, where NLL loss is adopted to alleviate degenerated advantages. Experiments show the effectiveness.

**Compliance With Llm Reviewing Policy:**

Affirmed.

**Final Justification:**

I think most of my concerns have been addressed, therefore, I will raise my score.

**Key Questions For Authors:**

1. Whether EchoClip update is adopted on all rollouts or advantages degenerated rollouts;
2. What if adopt the NLL optimization only on the hesitant step instead of all the prefix;

**Limitations:**

The authors haven't include limitations in their paper.

**Strengths And Weaknesses:**

**Strengths:**
1. The authors proposed to utilize the advantage-degenerated rollouts with identified hesitant point for NLL optimization;
2. The method seems easy to adapt for existing RL algorithms. Experiments show the effectiveness of the method on Qwen2.5 models;

**Weaknesses:**
1. More case analysis about the hesitant point identified by entropy should be presented;
2. The soundness of the EchoClip is doubted for NLL optimization, and the authors does not provide a formal analysis of why hesitant prefix identified by entropy given useful learning signals, and how does that change training behavior;
3. The comparison with other methods for handling advantage-degenerated rollouts should be presented;
4. It is suggested to provide the step entropy statistic to show the distribution, it is wonder how large this hesitant point differ from other steps;

---

> ### Author Rebuttal · Authors · 2026-03-30
>
> > **W1 & W4: Case Analysis and Entropy Statistics**
>
> First of all, we would like to clarify that EchoClip is **not** designed to extract only “hesitant points.” Rather, it identifies the step-level high-entropy region within a verified-success rollout, which may correspond to a genuinely critical but successful reasoning transition, e.g., an inspiring solution move. **A key clarification is that EchoClip should therefore not be understood as extracting a hesitant point such as a token like “wait”.** Our entropy signal is computed at the **step level** and is meant to localize low-probability but successful reasoning moves.
>
> Thus, the goal is to mine useful reasoning signal from successful rollouts rather than detect only hesitation markers as shown in **Appendix H**. In addition to the detailed four-rollout case study in **Appendix H** and entropy statistics in  **Figure 3a-b**, we will add more qualitative examples and more detailed plots/statistics in the revision.
>
> > **W2: Soundness of EchoClip for NLL optimization**
>
> First, every EchoClip is extracted from **a verified-success rollout**. This already ensures that the clip comes from a reasoning trajectory whose final outcome is correct, rather than from an arbitrary prefix. Second, **EchoClip is not chosen arbitrarily within that rollout**; it is further localized using a statistically motivated criterion, which is proven to be useful in the prior art [1, 2, 3] and our own analysis, namely that more informative reasoning trajectories tend to pass through higher-entropy regions.
>
> Figure 3a has shown that golden trajectories concentrate at higher entropy than self-generated rollouts, Figure 3b has shown that the more informative reasoning region exhibits higher entropy, and Figure 3c has shown that removing high-entropy steps hurts performance much more than removing low-entropy steps. Together, these results support that the interpretation that useful learning signal is concentrated in high-entropy successful regions. EchoClip therefore serves as a self-mined proxy for useful supervision: it is **both correct** (it comes from a verified-success rollout) and **informative** (it's localized to the same high-entropy regime that characterizes stronger supervision).
>
> Based on these evidences, it is not surprising that these segments provide additional training signal, whereas unfortunately all existing RLVR methods ignore it when advantage degeneration occurs. Instead, our EchoRL explicitly extracts and successfully reuses it through the auxiliary NLL term.
>
> > **W3: Comparison with Other Methods**
>
> Existing methods such as LUFFY [1] and UFT [2] operate along **an orthogonal axis** to EchoRL rather than serving as direct substitutes. Specifically, they strengthen training by introducing external teacher-provided trajectories, whereas EchoRL exploits supervision already latent in self-generated verified rollouts. Therefore, these methods are better viewed as **complementary mechanisms** rather than mutually exclusive alternatives, which is also reflected in Table 1.
>
> > **Q1: EchoClip on all rollouts or advantage-degenerated rollouts?**
>
> EchoClip update is applied to **all rollouts throughout training**, not only to explicitly advantage-degenerated rollouts. We use this unified design for simplicity and stability, instead of introducing a separate trigger for EchoRL. **We have also ablated restricted activation windows in Figure 8 (right)** and find that applying EchoRL across the full training run is a strong default.
>
> > **Q2: Full Prefix vs. Single Step**
>
> We find that supervising the **full successful prefix** is more effective than supervising only the isolated step. As clarified in our response to **W1 & W4**, the entropy signal is intended to localize a **high-value reasoning move**, not a superficial hesitation marker. Moreover, as noted in prior work on process supervision [3], the correctness of a reasoning step is inherently conditioned on its preceding context. Supervising the full prefix therefore provides a stronger and more coherent learning signal. The ablation above shows that **1-step supervision brings only marginal gain**, while **full-prefix supervision performs best** on both ID and OOD. This suggests that the useful signal lies not just in the pivot step itself, but in the successful reasoning context leading to it.
>
> |-|ID|OOD|
> |:-|:-:|:-:|
> |GRPO|44.5|55.6|
> |EchoRL(1-step)|44.9|56.2|
> |**EchoRL(Full)**|**47.0**|**58.4**|
>
> > **Limitation**
>
> EchoRL, like all RLVR methods, requires verifiable rewards and is inapplicable to domains without rule-based signals. We will add a dedicated Limitations section in the revision.
>
> We hope these clarifications thoroughly resolve your concerns and positively influence your final evaluation.
>
> [1]Learning to Reason under Off-Policy Guidance. NeurIPS 2025
>
> [2]UFT: Unifying Supervised and Reinforcement Fine-Tuning. NeurIPS 2025
>
> [3]Let's Verify Step by Step. ICLR 2024

---

> > ### Author Rebuttal · Reviewer_ViQp · 2026-04-04
> >
> > Thanks for authors response. I think some of my concerns have been addressed, however, the others still remain.
> > 1. Evidence of how prefix optimization affects the training behavior is still not provided;
> > 2. The case analysis of the extracted EchoClip is not provided in Appendix H, and it remains unclear about the extracted EchoClip;
> > 3. The entropy pattern in Fig. 3 appears to be evaluated on a single prompt, which raises concerns about its generality.
> >
> > Therefore, I will raise my score, but still can't recommend acceptance.

---

> > > ### Author Response · Authors · 2026-04-07
> > >
> > > We sincerely thank the reviewer for the careful follow-up and for **raising the score**. Below we clarify these points more directly.
> > >
> > > > **1. How prefix optimization changes training behavior.**
> > >
> > > Under GRPO, when all sampled rollouts for a prompt receive the same verified reward, the group-relative advantages collapse to zero, so the policy-gradient contribution from that prompt vanishes. **EchoRL changes this training behavior by adding an auxiliary likelihood term on the mined successful prefix.** Therefore, prompts that are silent under standard RLVR still produce dense gradients under EchoRL.
> > >
> > > Importantly, this is **not an arbitrary prefix**: the prefix ends at the highest-entropy successful step, so the update is concentrated on the very region where the model was most uncertain but still solved the problem correctly. In this sense, EchoRL does not simply “add imitation”; **it reactivates optimization on otherwise inactive successful groups and biases the update toward the pivotal correct decision region.** Empirically, this is consistent with our training-dynamics analysis: compared with GRPO, EchoRL maintains substantially higher step-level entropy during training and continues improving accuracy instead of collapsing early.
> > >
> > > > **2. What the extracted EchoClip actually is.**
> > >
> > > We apologize that our earlier appendix pointer was not clear enough. The intended case study is the quartic-polynomial example in Appendix H, where four rollouts all receive identical verified reward, but are not reasoning-equivalent: three follow a routine expansion route, while one uses the log-derivative trick. EchoRL identifies the highest-entropy step within the verified-success set and then takes the prefix ending exactly at that step as the EchoClip.
> > >
> > > **Concretely, the text below is the actual EchoClip selected from the Appendix H case study (the prefix extracted from the highest-entropy verified-success rollout among R1--R4):**
> > >
> > > > To find the critical points without full expansion, we can use the logarithmic derivative property.
> > > >
> > > > 1. **Center the Roots**: Shift the polynomial so the roots become symmetric around zero: $\{-3,-1,1,3\}$. Let the shifted variable be $y$.
> > > > 2. **Use Logarithmic Derivative**: For a polynomial $g(y)=\prod (y-r_i)$, the critical points satisfy $\frac{g'(y)}{g(y)}=\sum_i \frac{1}{y-r_i}=0$.
> > > > 3. **Set up the Equation**: Substitute the roots into the sum:
> > > >    $\frac{1}{y-3}+\frac{1}{y-1}+\frac{1}{y+1}+\frac{1}{y+3}=0$.
> > >
> > > **It is exactly the EchoClip mined from the four-rollout example in Appendix H**. Among the four verified-success rollouts, EchoRL selects the successful trajectory containing this high-entropy reasoning move and uses the prefix ending at this step as the auxiliary optimization target. This is exactly the kind of signal EchoRL is designed to preserve: not a superficial hesitation marker, but a compact, high-leverage reasoning move that resolves the problem through a more informative path.
> > >
> > > In the revision, we will make this appendix reference explicit and add a compact visual example that directly marks the selected EchoClip within the four Appendix H rollouts, so that the optimized target is completely unambiguous.
> > >
> > > > **3. Generality of the entropy evidence.**
> > >
> > > We agree that the current presentation may make Figure 3 look more local than intended. To clarify: **Figure 3b is a representative illustration**, but Figures 3a and 3c are dataset-level analyses rather than single-prompt anecdotes. Figure 3a compares entropy distributions of golden trajectories versus regular rollouts across prompts, and Figure 3c shows that removing high-entropy steps causes a much larger performance drop than removing low-entropy or random steps.
> > >
> > > Thus, our empirical basis is not a single example; **the single example is only used to make the mechanism concrete.** We further computed a direct statistic comparing the **selected high-entropy step against the remaining steps:**
> > >
> > > | Step type                  | Mean entropy |
> > > | :------------------------- | :----------- |
> > > | Selected high-entropy step | 2.23         |
> > > | Other steps                | 1.84         |
> > >
> > > This quantitative gap further supports that **EchoClip is not picking an arbitrary prefix location,** but is indeed anchored at a distinctly higher-uncertainty region. In the revision, we will add this statistic to the paper and revise the surrounding text/caption to distinguish more explicitly between illustrative case analysis and dataset-level evidence.
> > >
> > > ---
> > >
> > > Overall, our claim is not that any prefix-level NLL is helpful. Rather, the useful signal comes from **verified-success prefixes anchored at the highest-entropy correct step**, which provide a concrete way to convert advantage-degenerated but still informative groups into active training signal. We appreciate the reviewer’s push on this point and will make this causal chain substantially clearer in the revision.

---

### Official Review · Reviewer_Y2hw · 2026-03-12

**Soundness:** 3
**Presentation:** 4
**Significance:** 2
**Originality:** 3
**Overall Recommendation:** 4
**Confidence:** 3

**Summary:**

This paper introduces EchoRL: an extension of GRPO-like families for reinforcement learning with verifiable rewards. EchoRL aims to address "advantage degeneration"; a term introduced for the problem in GRPO where, after some training, many prompts in the dataset are easily solved causing all rollouts to arrive at a correct answer. In turn, advantages tend to 0 resulting in a collapse of learning signal. The authors note that, even when final answers are all equally correct, the reasoning for some answers may still be preferred over others and should thus be reinforced differently.

Empirically, the authors identify that better reasoning logic usually coincides with rollouts containing high-entropy spikes in some token chains. To this end, they propose EchoRL which aims to find these spikes of high-entropy chains and reinforce them.
The authors argue that EchoRL can easily be integrated into GRPO-like algorithms and perform experiments with GRPO+EchoRL and DAPO+EchoRL. They find that EchoRL increases performance across the board on various benchmarks, at a minimal computational cost.

**Compliance With Llm Reviewing Policy:**

Affirmed.

**Final Justification:**

Based on the rebuttal concerning my two key questions. I slightly increase my score.

**Key Questions For Authors:**

1.	I would most likely alter my score if you could make the connection and/or comparison to simply adding an entropy bonus term to the loss. This appears to be defaulted to 0.0 in the Verl implementation that you have based your work on. I can see that this default is likely set for a reason, but intuitively the effect of this parameter could be somewhat similar. I personally do not perform any LLM finetuning research, so take this all with a grain of salt in case I am wildly off here.
2.	Could you expand on contribution 1? As it stands, I feel like the contribution is coining a term for a known problem. See the strengths and weaknesses for my concerns.

**Limitations:**

The authors have not listed any limitations. Instead, the work is coined as a "free lunch" module.

**Strengths And Weaknesses:**

- The ideas of the paper are very well presented, both in clear writing as well as in excelent visualization of ideas and results.
- EchoRL seems like a promising and solution with wide applicability and strong perfomance improvements.
- The idea of enforcing high-entropy rollouts is seems sound as it is well backed-up by emperical experiments and is properly explained and visualized.
- However, this finding may not be all too significant and original as prior work had already pinpointed to the issue of entropy collapse and advantages trending to 0 (resulting in a diminishing learning signal). As such, I have huge issues with contribution 1, which lists the identification of "advantage degeneration" as a clear contribution. All the while, the related works lists papers that have tackled the same problem and therefore must have identified it. Other than coining a new name, I do not fully see the contribution here.
- At its core, EchoRL attempts to maximize rewards while producing high entropy outputs. This practice is quite common in RL and therefor EchoRL appears to be a bit more involved than perhaps necessary. A comparison to simply adding a bonus entropy term to loss function is not drawn.
- No limitations mentioned.

Minor:
- Figure 3a might be better visualized with the same limits on the y-axis.

---

> ### Author Rebuttal · Authors · 2026-03-30
>
> ## Response
>
> We sincerely thank the reviewer for recognizing our clear presentation and the strong performance improvements of EchoRL.
>
> > **Weakness & Question 1: Could you make the connection and/or comparison to simply adding an entropy bonus term to the loss?**
>
> Let us try to explain why simply adding an entropy bonus term won’t achieve the goal.
>
> In fact, in mainstream RLVR frameworks (including `verl`, which we build upon),  **An entropy bonus is already enabled** by default (specifically, `actor_rollout_ref.actor.entropy_coeff = 0.001`). The standard actor loss being optimized is exactly:
>
> $$
> L_{base}=
> -\mathbb{E}\!\left[
> \min\!\left(
> \frac{\pi_\theta}{\pi_{old}}\hat A,\;
> clip\!\left(\frac{\pi_\theta}{\pi_{old}},1-\epsilon,1+\epsilon\right)\hat A
> \right)
> \right]
> -c_{entropy}H(\pi_\theta)
> $$
>
> Despite this built-in entropy constraint, **models still suffer from severe advantage degeneration as shown in Table 1 and Figure 1**. This shows that simply adding a scalar entropy bonus does not work.
>
> **EchoRL is a complementary and distinct innovation on top of these standard implementations.** We retain all default framework parameters and formulate our objective as an addition to the base loss:
>
> $$
> L_{EchoRL}=L_{base}+\lambda L_{EchoClip}
> $$
>
> , where the entropy bonus is **already** included in $L_{base}$, with coefficient $c_{entropy}=0.001$ (see Appendix).
>
> The problem of using the entropy regularizer is that when advantage degeneration occurs, simply encouraging higher-entropy outputs does not tell the model which part of which successful rollout is actually worth reinforcing. EchoRL exactly fixes this problem by telling where the specific high-value segments are and how to utilize them. In this sense, our method is not “entropy bonus in a more complicated form,” but a different training signal built upon with the precise information that vanilla entropy regularization cannot localize.
>
> > **Weakness & Question 2: Could you expand on contribution 1? As it stands, I feel like the contribution is coining a term for a known problem.**
>
> We fully agree with the reviewer that advantage degeneration is not our discovery. Advantage degeneration is exactly the bottleneck RVLR suffers, which is also the problem our work aims to address. To solve the problem, our contributions are 1) providing a method that can **identify reusable supervision** inside self-generated successful rollouts and 2) utilizing the identified in-band signal to **further improve the model training** performance. The key difficulty is that, under advantage degeneration, verified-success rollouts receive the same reward, while the reward itself is blind to reasoning-path quality and thus cannot reveal which part of the rollout is actually worth reinforcing. The present EchoRL closes this gap and provides a lightweight but effective solution that has never been done in prior art. We will fine-tune the statement to clarify our actual contribution without any possible misunderstanding or overclaiming.
>
> > **Weakness: No limitations mentioned.**
>
> **EchoRL shares the same inherent scope boundary as all mainstream RLVR methods**: it strictly requires verifiable training signals. The RLVR category, including EchoRL, is naturally not designed for domains without rule-based rewards (e.g., open-ended creative writing), and thus cannot be applied to training in such domains. We will add a dedicated Limitations section in the revised manuscript to explicitly and carefully discuss this paradigm boundary.
>
> We hope these clarifications thoroughly resolve your concerns and positively influence your final evaluation.

---

> > ### Author Rebuttal · Reviewer_Y2hw · 2026-04-04
> >
> > A thank you to the authors for their response.
> >
> > 1. In contrast to your response, I believe in Verl, the default entropy_coeff is currently disabled (see [here](https://github.com/verl-project/verl/blob/d5c5daa84290b445f071fb812a1731a7325f350f/verl/trainer/config/actor/actor.yaml#L89) and [here](https://github.com/verl-project/verl/commit/aab01764af4777859be66d3a3d3beb53a1b24a55)). Nevertheless, it is clearly enabled in your experiments, which is what matters and I therefore stand corrected. The author's response does not fully satisfy the "why" of how the EchoRL additive loss would arrive at a different solution than simply adding an entropy bonus, but as the entropy bonus is included in the baselines, I can see that it empirically does (even though wider experimental results on this would have been preferred). Furthermore, after reexamining the text, I can see the reasoning. Overall, I am reasonably satisfied with the response.
> >
> > 2. "providing a method that can identify reusable supervision" seems a lot better formulated compared to "We identify a key deficiency common in RLVR methods". Although the rest of your original contribution still stands (formulating how prior methods dealt with the problem). I am satisfied with the response.
> >
> > EchoRL is still claimed to be a free lunch module, which I think remains a strong claim. Especially, considering limitations are acknowledged and the paper includes results in which EchoRL performs worse compared to baselines.
> >
> > Based on the rebuttal, I will increase my score.

---

> > > ### Author Response · Authors · 2026-04-07
> > >
> > > We sincerely thank the reviewer for the careful reading, constructive follow-up, and for **increasing the score after the rebuttal**. We are especially encouraged that the reviewer found our revised explanation of Contribution 1 substantially better and was reasonably satisfied with the clarification regarding the entropy-bonus question. Overall, we believe the main concerns **have been largely resolved.** We summarize the key points below.
> > >
> > > > **1. EchoRL is not equivalent to simply adding an entropy bonus**
> > >
> > > We thank the reviewer for pushing us to clarify this point more precisely. The key distinction is that an entropy bonus and EchoRL provide **fundamentally different optimization signals**. A standard entropy-regularized RLVR objective can be written as:
> > >
> > > $$
> > > L_{base}=
> > > -\mathbb{E}\!\left[
> > > \min\!\left(
> > > \frac{\pi_\theta}{\pi_{old}}\hat A,\;
> > > clip\!\left(\frac{\pi_\theta}{\pi_{old}},1-\epsilon,1+\epsilon\right)\hat A
> > > \right)
> > > \right]
> > > -c_{entropy}H(\pi_\theta)
> > > $$
> > >
> > > Where the entropy term is defined as:
> > >
> > > $$
> > > H(\pi_{\theta}(\cdot | s_t)) = - \sum_{y} \pi_{\theta}(y | s_t) \log \pi_{\theta}(y | s_t)
> > > $$
> > >
> > > This term is **state-wise and non-directional**: it encourages a flatter next-token distribution at visited states, but it does not specify which successful trajectory, or which segment within it, should be reinforced. By contrast, EchoRL uses a success-conditioned auxiliary objective:
> > >
> > > $$
> > > L_{EchoRL}(\theta) = - \sum_{t \in EchoClip} \log \pi_{\theta}(o_t^{echo} | q, o_{<t}^{echo})
> > > $$
> > >
> > > The total objective is then:
> > >
> > > $$
> > > L(\theta) = L_{RLVR}(\theta) + \lambda L_{EchoRL}(\theta)
> > > $$
> > >
> > > This signal is **success-conditioned, segment-localized, and token-directional**. More concretely, EchoRL (1) restricts attention to verified-success rollouts, (2) identifies the highest-entropy critical step, and (3) reinforces the corresponding successful prefix through a likelihood objective.
> > >
> > > Therefore, **an entropy bonus mainly promotes exploration, whereas EchoRL supplies targeted credit assignment when group-relative rewards become non-informative**. Importantly, in our experiments, the baseline already includes entropy regularization, yet advantage degeneration still appears and EchoRL still yields clear gains.
> > >
> > > > **2. Contribution 1 is not the discovery of the pathology itself**
> > >
> > > We thank the reviewer for recognizing this clarification. We will revise the paper to avoid any overstatement on this point.
> > >
> > > > **3. We agree that “free lunch” is too strong, and will revise the wording and add limitations**
> > >
> > > We also thank the reviewer for pointing this out. We agree that “free lunch” is too strong a phrase, and we will revise it. What we intended to claim is **low additional system overhead,** not universal monotonic gains on every benchmark. EchoRL reuses the same rollouts and the same forward/backward pass, so its computational overhead is negligible.
> > >
> > > **We will therefore replace “free lunch” with “lightweight, low-overhead plug-and-play module” in the revision**. We will also add a dedicated Limitations section to make the scope explicit.
> > >
> > > > **4. Overall**
> > >
> > > We are grateful that the reviewer **found the rebuttal helpful and raised the score accordingly**. We therefore believe the central concerns have been **substantially resolved.** The remaining points are mainly about wording calibration and explicit discussion of scope and limitations. We are fully committed to addressing both in the revision.

---

### Official Review · Reviewer_h4sq · 2026-03-13

**Soundness:** 2
**Presentation:** 3
**Significance:** 2
**Originality:** 3
**Overall Recommendation:** 4
**Confidence:** 3

**Summary:**

This paper proposes EchoRL that additionally uses prefix reasoning steps (EchoClip) as supervised signals to address the advantage degeneration problem of reinforcement learning with verifiable rewards (RLVR). This paper evaluates EchoRL across nine benchmarks (e.g., AIME24, AIME25, MATH-500, etc) on five LLM backbones (i.e., 1.5B ~ 8B). Experiment results show that EchoRL can further increase the performance of group-based reinforcement learning methods such as GRPO and DAPO.

**Compliance With Llm Reviewing Policy:**

Affirmed.

**Final Justification:**

This paper proposes EchoRL that uses EchoClip (i.e., a prefix partial reasoning trajectory with the highest entropy step) as a supervised learning signal in addition to a group-based policy optimization loss, aiming to address the advantage degeneration problem of reinforcement learning with verifiable rewards (RLVR). In my review, I raised one weak point (i.e., the inconsistency of the performance improvements across math benchmarks) and two questions (i.e., (1) the advantage of using EchoClip (i.e., a partial trajectory) over a whole trajectory, and (2) any side effects of using EchoClip as supervised learning signals). The authors provided thoughtful responses, and the responses largely resolved my concerns and questions. Therefore, I maintain my initial positive score (i.e., 4: Weak accept).

**Key Questions For Authors:**

- Q1. What are advantages of using EchoClip (i.e., a partial trajectory with the highest entropy step) over using the hole trajectory if the trajectory is correct?
- Q2. Are there any side effects of additionally using EchoClip as training signals? For example, is there any tendency of overfitting to some typical prefixes of reasoning trajectories?

**Limitations:**

yes

**Strengths And Weaknesses:**

Some strengths of this paper can be summarized as follows:
- S1. First of all, this paper is clearly written and well organized.
- S2. This paper identify advantage degeneration, a key problem that hinders the performance improvements in RLVR.
- S3. The proposed method EchoRL seems very practical.
- S4. This paper provides comprehensive experiment results across nine benchmarks (e.g., AIME24, AIME25, MATH-500, etc) on five LLM backbones (i.e., 1.5B ~ 8B).

Some weaknesses of this paper can be summarized as follows:
- W1. One of weaknesses of this paper may be the inconsistency of the performance improvements across math benchmarks. For example, according to Table 1, the performance of GRPO+EchoRL decreases from 25.8 to 24.9 on AIME24, while it increases from 16.4 to 22.3 on AIME25.

---

> ### Author Rebuttal · Authors · 2026-03-30
>
> ## Response
>
> We sincerely appreciate your positive review and your recognition of our clearly written paper, our identification of the advantage degeneration problem, and our practical method supported by comprehensive experiments.
>
> > **Weakness 1: Inconsistency of performance improvements across math benchmarks**
>
> We agree that the performance is not strictly monotonic on every individual math benchmark. However, a fairer assessment is the aggregated trend across tasks, models, and independent runs. Specifically, across the 4 backbones reported in Table 1 and Appendix A and the 6 in-distribution math benchmarks, **EchoRL improves 22 out of 24 GRPO-based model-task combinations**, with only 2 slight decreases. Meanwhile, on the main Qwen2.5-Math-7B setting, the overall ID average still improves from 44.5 to 47.0, and Appendix B further shows stable multi-run gains. For small, high-difficulty benchmarks such as AIME24/AIME25, actually, the fluctuation is expected but this is not a reliable indicator. This is expected because AIME24/AIME25 are **small-scale and extremely challenging** benchmarks, so their reported performance is naturally subject to higher variance: even a difference of one or two solved problems can lead to a visible score fluctuation. In addition, under such hard regimes, the number of effective successful rollouts is sparse, making overall single-benchmark results more sensitive to sampling and decoding randomness than the aggregated trend across tasks and runs. **From a more general/multi-dimensional perspective, in our humble opinion, the benefit of EchoRL is highly consistent.**
>
> > **Question 1: What are advantages of using EchoClip over using the whole trajectory if the trajectory is correct?**
>
> We understood the motivation behind the reviewer’s proposal. However, **the usable learning signal only concentrates in a small critical segment**, and effective supervision requires isolating that segment rather than imitating the whole rollout. The reason is that the whole rollout mixes the pivotal reasoning step with many routine, low-information tokens, which will dilute the learning signal. Hence, the full correct trajectory is not an equally useful supervision target, meaning not helpful for further improving RVRL.
>
> Conversely, the present EchoClip works because it accurately isolates the part that actually carries the improvement signal. **Figure 3 provides supporting evidences**: high-entropy steps mark these usable signals, as removing them hurts performance much more than removing low-entropy or random steps.
>
> Nevertheless, efficiently identifying the critical segment is not a trivial task, because under advantage degeneration all verified-success rollouts receive the same reward, while the reward is blind to reasoning-path quality and therefore does not tell us which part of which rollout is actually worth reinforcing. To this end, the proposed EchoClip algorithm is necessary because it exactly enables EchoRL to solve this non-trivial extraction problem.
>
> **In fact, we have compared the two approaches in Appendix C / Figure 8.** We directly compare using the full verified rollout as the auxiliary target against using EchoClip. EchoClip consistently performs better: for GRPO, full-rollout supervision reaches 45.2/56.1 ID/OOD average, whereas EchoClip reaches 47.0/58.4; for DAPO, the corresponding numbers are 48.3/58.0 vs. 49.9/60.9.
>
> These results concludes that EchoClip yields the strongest ID/OOD gains, supporting our claim that the useful signal must be localized rather than applied to the whole correct trajectory.
>
> > **Question 2: Are there any side effects of additionally using EchoClip as training signals?**
>
> This is a valid concern. **Empirically, we do not observe evidence that EchoClip causes the model to overfit to stereotypical reasoning prefixes.** In particular, the paper already includes a targeted analysis of length/stylistic effects: Figure 7 shows that the mean response length of GRPO+EchoRL closely tracks vanilla GRPO throughout training and is sometimes even slightly shorter, suggesting that the gains are not explained by simply copying longer or more typical prefixes. The appendix also states this explicitly as evidence against a trivial length-driven or stylistic overfitting effect.
>
> We hope these clarifications will further strengthen your already positive evaluation.

---

> > ### Author Rebuttal · Reviewer_h4sq · 2026-04-04
> >
> > I would like to thank the authors for providing thoughtful responses to my comments and questions. The answers provided by the authors largely resolved my questions. Therefore, I maintain my initial positive score.

---

> > > ### Author Response · Authors · 2026-04-07
> > >
> > > We sincerely thank the reviewer for the positive assessment, the constructive questions, and the thoughtful follow-up after the rebuttal. We are especially encouraged that the reviewer marked the concerns as **fully resolved** and maintained a **positive** recommendation. We also greatly appreciate the reviewer’s recognition of the paper’s clarity, practical value, identification of the advantage degeneration problem, and the breadth of the experimental evaluation. In our view, this indicates that the main concerns regarding benchmark variability, the motivation for EchoClip over full-trajectory supervision, and possible side effects such as prefix overfitting have been adequately addressed.

---

### Decision · Program_Chairs · 2026-04-30

**Decision:**

Accept (regular)

**Comment:**

The reviewers were broadly positive about the paper’s practical value, clarity, and empirical strength, and the final ratings converged to 4/4/4/4. Reviewers h4sq and beb5 maintained positive scores throughout, while Reviewers Y2hw and ViQp both raised their scores after rebuttal.

The main concerns were about the distinction from entropy regularization (Y2hw), the soundness and analysis of EchoClip (ViQp), benchmark consistency and possible side effects (h4sq), and design/scaling details (beb5). The rebuttal addressed these points with clearer framing of the contribution, stronger mechanistic explanation, additional empirical evidence including larger-scale results, and explicit revision commitments on wording and limitations.

Overall, the paper presents a clear and practical method for an important RLVR training bottleneck, with solid empirical support and reviewer consensus after rebuttal. I recommend accept.